# Transforming Visual Classifiers for Zero-Shot Text-Based Interpretability

## Abstract

Visual classifiers offer high-dimensional feature representations that are challenging to interpret and analyze. Text, in contrast, provides a more expressive and human-friendly interpretable medium for understanding and analyzing model behavior. We propose a simple, yet powerful method for reformulating any visual classifier so that it can be accessed with open-set text queries without compromising its original performance. Our approach is label-free, efficient, and preserves the underlying classifier's distribution and reasoning processes. We thus unlock several text-based interpretability applications for any classifier. We apply our method on 40 visual classifiers and demonstrate two primary applications: 1) building both label-free and zero-shot concept bottleneck models and therefore converting *any* classifier to be inherently-interpretable and 2) zero-shot decoding of visual features into natural language. In both applications, we achieve state-of-the-art results, greatly outperforming existing works. Our method enables text approaches for interpreting visual classifiers.[1]

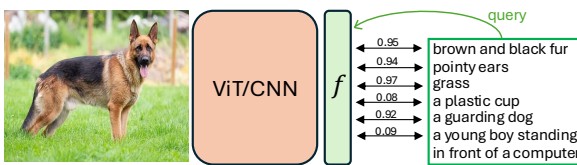

Figure 1. We formulate *any* classifier to allow querying it and measuring its association with any text, while maintaining its performance and distribution. $f$ is the feature vector of an image

## 1. Introduction

Visual classifiers inherently and purely provide visual signals, which are high-dimensional dense vector representations of images that are hard to interpret and dissect, limiting attribution maps as the main medium for interpretation. Attribution maps provide general and high-level information, indicating where in the image the most important region is for the prediction. Nevertheless, text is significantly more expressive, human-friendly, specific, and unlocks a variety of other interpretability applications that require language (Tewel et al., 2021; Koh et al., 2020; Oikarinen & Weng, 2023; Song et al., 2022; Minderer et al., 2022). Therefore, a question that arises is *how can we talk or access those visual classifiers via open-set text?* Performing this allows us to have a text interface that can be used to query visual classifiers. So far, this has only been achieved with generic vision-language models such as CLIP-based (Radford et al.,

2021), which learn a shared vision-language space and thus enables any text to query the visual model directly. However, the deep learning field extends far beyond CLIP. For example, industries—being the primary end-users of such models—often rely on custom visual models tailored to their specific needs. This creates an application bottleneck: any task involving both visual and textual interpretations becomes restricted to CLIP or similar approaches, limiting flexibility, forcing reliance on such models, and inheriting their limitations.

In this work, we lift this restriction by proposing a method to reformulate any visual classifier such that it can be accessed via open-set text (Figure 1). Our method is characterized by four important properties: First, it is *efficient*; it is inexpensive to train and can be performed on any standard hardware, regardless of the size of the original classifier. Secondly, it is *label-free*, no labels are required to achieve this formulation. Thirdly, it is *faithful* to the classifier and preserves its underlying distribution and reasoning process. Finally, our method is applicable to *any* vision architecture, whether convolutional-based, transformer-based or hybrid.

A visual classifier assigns an image to a specific category from a predefined set of discrete class labels. In ImageNet-1K (Deng et al., 2009), there are a total of $1,000$ labels. Originally, these discrete class labels correspond to class names in text format. For example, in ImageNet-trained models, the discrete label 1 corresponds to the class *goldfish*. These classes are typically discretized to facilitate training with cross-entropy. However, when the textual class names are embedded into vector representations (*e.g,* using a text encoder or word embedding model), they provide semantic information. Specifically, these embeddings also semantically encode words that are close to the class name within the text embedding space. For the "goldfish" example, the

---

[1]Code will be available

closest words could be "freshwater", "fins" and "orange". Our method learns to map images into this text embedding space, thus associating both the class name *and its surrounding semantic space* with the image. This is accomplished through a trainable MLP that projects the visual features into the text embedding space, and is trained by aligning its output distribution across all classes with the original classifier's output distribution, while keeping both the visual and textual encoders frozen. By using solely the class names without any supplementary information, we can learn a semantically meaningful image-text space. This allows us to query the visual classifier with any text query beyond the class names.

We demonstrate the effectiveness of our method with two primary applications. 1) We build label-free concept bottleneck models for any classifier (not being restricted to CLIP) while obtaining the linear probe classifier in a zero-shot manner (we do not train a linear probe on the concept activations), also ensuring that the new linear classifier remains faithful to the original classifier. 2) We show that our method can decode visual features of any classifier into natural language in a zero-shot manner, without requiring training on image-text data such as image-caption pairs, while also showing the generalization of our method to datasets beyond ImageNet.

In summary, our contributions are as follows:

- We propose an efficient, label-free method to reconfigure any visual classifier such that it can be queried with open-set text queries, while also being faithful to the classifier.

- We demonstrate the effectiveness of our method with **40 different architectures** through two applications: label-free, faithful zero-shot Concept Bottleneck Models (CBMs) for any visual classifier, and zero-shot decoding of visual features into natural language.

- Our method sets new state-of-the-art results outperforming existing works, including those which use CLIP as the visual encoder, although being trained with $400\times$ less images and $400,000\times$ less captions, thus *efficiently* learning *faithful* text interfaces to interpret any visual classifier.

## 2. Related Work

While CLIP is the predominant approach to interpret vision models through language (Oikarinen & Weng, 2023), there exist works that try to decode visual features of models beyond CLIP. DeVIL (Dani et al., 2023) trains an autoregressive text generator to map visual features from different layers of a classifier into image captions, leveraging annotated image-caption pairs as ground-truth data. Similarly,

Natural Language Explanations (NLEs) (Park et al., 2018; Kayser et al., 2021; Sammani et al., 2022) use annotated textual explanations in place of conventional captions. Notably, all these works (1) rely on annotated datasets and (2) explicitly train the generated text to align with what annotators want the visual features to describe, and are therefore not faithful to the classifier. ZS-A2T (Salewski et al., 2023) maps attention maps into natural language in a zero-shot manner using LLMs, but is constrained to vision-language models trained to learn a shared vision-language embedding space (e.g., through a contrastive objective), and thus cannot be applied to visual classifiers. Our work on the other hand decodes visual features of any classifier in a zero-shot manner without requiring any annotated data, while also maintaining faithfulness to the original classifier.

Text-to-Concept (T2C) (Moayeri et al., 2023) is the closest related work to ours, where a linear layer is trained to map image features of any classifier into the CLIP vision encoder space, such that they can be interpreted via text using the CLIP text encoder. Since the linear layer maps features to the CLIP space, T2C is strongly biased towards interpreting the CLIP model rather than the original classifier. As the ground-truth data for training the linear mapper are the features extracted from the CLIP vision encoder, this method is approximate to using the CLIP vision encoder directly to encode the image. Moreover, T2C relies on CLIP supervision. In contrast, our approach is label-free and our mapping function is explicitly trained to preserve the classifier's original distribution, ensuring faithfulness to it.

## 3. Method

Consider an image $I$ and a visual classifier $F$ composed of a visual feature extractor $F_v$ and a linear classifier $W$. Note that $F$ can be of any architecture. $F_v$ embeds $I$ into an $n-$dimensional feature vector $f \in \mathbb{R}^n$. That is, $f = F_v(I)$. The linear classifier $W \in \mathbb{R}^{n \times K}$ takes $f$ as input and outputs a probability distribution $o$ for the image across $K$ classes. That is, $o = \text{softmax}(f.W) \in \mathbb{R}^K$. For ImageNet-1K, $K = 1000$. Consider also any off-the-shelf text encoder $T$ which takes in an input text $l$ and embeds it into a $m-$dimensional vector representation $u \in \mathbb{R}^m$. That is, $u = T(l)$. Note that $u$ and $f$ are not in the same space and can have a different number of dimensions, so we cannot query $f$ with the text $l$.

We propose to learn a simple light-weight MLP mapping function which projects the visual features $f$ into the text embedding space of $T$, resulting in a new vector $\tilde{f}$. That is, $\tilde{f} = \text{MLP}(f)$, where $\tilde{f} \in \mathbb{R}^m$. Note that the visual encoder $F_v$, the linear classifier $W$, and the text encoder $T$ are all frozen; only the MLP is trainable, making our method *efficient*. We then take the textual class names of the $K$ classes, and convert each into a text prompt $l^p$, repre-

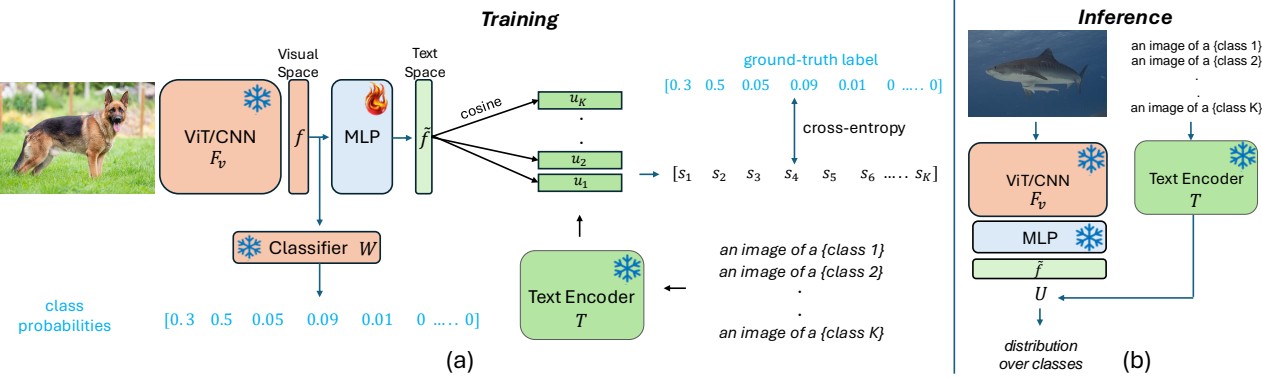

*Figure 2.* An overview of our method for reformulating visual classifiers. (a) The process of training the MLP mapping between vision and text space. (b) The process of inference with adapted visual classifier. The text encoder acts as a linear classifier weight generator in our re-formulated classification process. ❄ indicates that the module is frozen, while 🔥 indicates trainable.

sented as: "an image of a {class}" where {class} represents the class name in text format. This results in $K$ textual prompts, each of which is encoded with $T$: $u_i = T(l_i^p)$, $\forall i = 1, \ldots, K$. Stacking all the encoded prompts, we get a matrix $U \in \mathbb{R}^{K \times m}$. We then calculate the cosine similarity[2] between each $u_i$ and the visual features $f$: $s_i = \tilde{f}.u_i$. Equivalently, this can be performed as a single matrix multiplication: $S = \tilde{f}.U^T$, where $S \in \mathbb{R}^K$ represents the cosine similarity scores between the visual features and every text prompt $l_i^p$ representing a class.

The most straightforward approach to training the MLP mapper is to leverage the ground-truth labels from the dataset, aligning $S$ with the ground-truth distribution. However, this approach violates two key properties: (1) it necessitates annotated data, and (2) re-training the classifier alters its original distribution $o$, thereby changing the reasoning process of the classifier (i.e., how it maps visual features to class probabilities and makes predictions). Notably, the original soft probability distribution $o$ is a function of the linear classifier $W$, so $W$ cannot be ignored. We instead propose to align $S$ to the original distribution $o$ through the cross-entropy loss. For a single sample, the loss is given by:

$$L = -\sum_{i=1}^{K} o_i \log \left( \frac{e^{s_i}}{\sum_{j=1}^{K} e^{s_j}} \right) . \quad (1)$$

This task can be viewed as a knowledge distillation problem, except that we do not distill the knowledge of a bigger teacher model to a smaller student model, but distill the distribution of the original model to a reformulated way of classification. This makes our approach label-free and preserves the distribution and reasoning process of the classifier $F$. We provide a PyTorch-like pseudocode of our training approach in Listing 1, and illustrate it in Figure 2(a). It is

____________
[2]in the rest of this paper, we will omit the unit norm in cosine similarity to reduce clutter, and represent it with the dot product

important to note that we only use the class name to formulate the textual prompt $l^p$, and no other supplementary information such as class descriptions or hierarchies (see Section F in the appendix for more information).

After training, the projected visual features and the text encoder features lie in the same space, and we can therefore query the visual features with any text by finding the alignment score between the encoded text and the visual features. In the case of image classification, the text queries remain the class prompts, and encoding them with the text encoder $T$ is equivalent to generating the weights of a linear classifier for the newly formulated classification task, defined as $\arg\max(\tilde{f}.U^T)$. This is shown in Figure 2(b). We show later that this weight generating process is important to build faithful concept bottleneck models.

```
# text_feats: textual features of class names from a frozen
    sentence encoder, shape (num_classes, text_dim)
# classifier: linear classifier weights of a frozen
    vision_encoder, shape (visual_dim, num_classes)
# mlp: trainable MLP from visual_dim -> text_dim
# images: batch of B images, shape (N, 3, height, width)

visual_feats = vision_encoder(images) # (N, visual_dim)
logits = visual_feats @ classifier # (N, num_classes)
original_dist = softmax(logits, dim=-1) # (N, num_classes)

mapped_feats = mlp(visual_feats) # (N, text_dim)
mapped_feats = l2_norm(mapped_feats) # (N, text_dim)
text_feats = l2_norm(text_feats) # (N, text_dim)

pred_logits = mapped_feats @ text_feats.T # (N, num_classes)
pred_dist = softmax(pred_logits, dim=-1) # (N, num_classes)

# cross entropy with original model's soft distribution
loss = -(original_dist * log(pred_dist)).sum(dim=1).mean()
loss.backward() # only mapper parameters are updated
```

*Listing 1.* PyTorch-like pseudocode for our method

## 4. Applications

In this section, we show how our method can be used to build text-based interpretability applications. At application, all model components (including the MLP) are frozen.

## 4.1. Zero-Shot Concept Bottleneck Models

Concept Bottleneck Models (CBMs) (Koh et al., 2020) are a class of inherently interpretable models and have recently attracted significant attention. They consist of two steps: (1) concept discovery, followed by (2) concepts-to-predictions. In (1), the dense output features of a visual encoder are first mapped to textual concepts (e.g., words or short descriptions of objects) each with a score which represents the concept activation to the image. In (2), a linear classifier $W^{con}$ is trained on top of these concept activations to predict the class such that a prediction can be interpreted as a linear sum of interpretable concepts rather than a linear sum of dense features. Recently, the Label-Free CBMs (LF-CBMs) family uses CLIP to perform the concept discovery step without annotated image-concept data, by either using CLIP as a model to provide ground-truth image-concept similarities to train a layer which performs concept discovery (Oikarinen et al., 2023), or by querying the image features from a set of predefined concepts within the CLIP space (Yang et al., 2022; Panousis et al., 2023) and using the cosine similarity scores between the concepts and the image as concept activations. However, there are major issues in the current literature of CBMs. Firstly, LF-CBMs are restricted to CLIP-based models, as the concept discovery step would otherwise not be possible since it necessitates an image-text similarity model. Secondly, all CBM approaches to date require training a linear probe on the concept activations which 1) hinders an on-the-fly ready-to-deploy CBMs, and 2) disconnects $W^{con}$ from the original feature-based classifier $U$, since $W^{con}$ is trained from scratch in a different space using ground-truth data and is hence not faithful to the original model, and 3) requires training a new $W^{con}$ classifier for every different set of concepts, hindering flexibility to the choice of concepts.

Our method allows us to solve all the aforementioned issues and formulate zero-shot CBMs for any classifier. We remind readers from Section 3 that $U \in \mathbb{R}^{K \times m}$ represents the classification weights of the newly formulated classifier (the output of the text encoder $T$ for the class prompts). We assume that we are given a large set of predefined textual concepts, denoted as $\mathcal{Z}$, and with cardinality $|\mathcal{Z}| = Z$. Following other works (Rao et al., 2024), and without loss of generality, we use the $Z = 20K$ most common words in English[3] as our concept set. These are general concepts that represent world knowledge and are not tailored towards any specific dataset. We use the same text encoder $T$ that generates the linear classifier $U$ to generate concept embeddings, by feeding each concept $z_i \in \mathcal{Z}$ to the text encoder $T$ to generate a concept embedding $c_i$. That is, $c_i = T(z_i)$, $\forall i = 1, \ldots, Z$. By performing this for all $Z$ concepts, we obtain a concept embedding matrix $C \in \mathbb{R}^{Z \times m}$. For an im-

---

[3] https://github.com/first20hours/google-10000-english

age $I$, we extract its visual features $f$ and use MLP to map them to $\tilde{f}$ which now lies in the text embedding space. That is, $\tilde{f} = \text{MLP}(f)$, and $\tilde{f} \in \mathbb{R}^m$. Since $C$ and the mapped visual features $\tilde{f}$ are now in the same space, we can query $\tilde{f}$ to find which concepts it responds to. That is, we perform concept discovery using the cosine similarity between $\tilde{f}$ and $C$. The concept activations are obtained by $\tilde{f}.C \in \mathbb{R}^Z$ and represent the activation score for each of the $Z$ concepts. We provide an illustration in Figure 3.

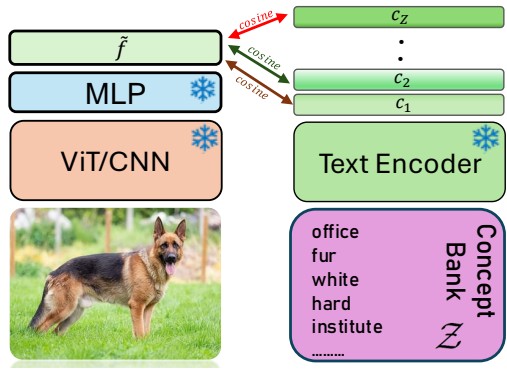

*Figure 3.* Our suggested approach to concept discovery in CBMs.

We now show how to build the classifier $W^{con}$ that takes in the concept activations and outputs a distribution $S_{cn}$ over classes. We build $W^{con}$ in a zero-shot manner. Here, *zero-shot* indicates that no training is required to map concept activations to classes. Recall that both $U$ and $C$ are outputs of the text encoder $T$, and they are already in the same space. Therefore, we can build the weights of the classifier $W^{con}$ with a text-to-text search between the concepts and the class name. Specifically, we calculate the cosine similarity between the concept embeddings $C$ and the linear classifier $U$ to obtain the new weights for $W^{con}$. That is, we perform $C \cdot U^T \in \mathbb{R}^{Z \times K}$. Therefore, the weights of $W^{con}$ represent how similar the class name is to all the concepts. In total, the output distribution $S_{cn}$ of the CBM is defined as:

$$S_{cn} = \underbrace{(\tilde{f} \cdot C^T)}_{\text{concept discovery}} \cdot \underbrace{(C \cdot U^T)}_{\text{concept-to-class}} = \tilde{f} \cdot \underbrace{C^T C}_{\text{gram matrix}} \cdot U^T . \quad (2)$$

By observing Eq. 2, we observe that we scale the linear feature-based classifier $U$ by the gram matrix of concepts ($C^T C \in \mathbb{R}^{m \times m}$). Notably, if the gram matrix is identity ($C^T C = I$), we get back our original feature-based classifier given by $\tilde{f}.U^T$. Therefore, to convert any classifier to a CBM, all we have to do is plug in the gram matrix in-between, making it a convenient way to directly switch to an inherently interpretable model. Eq. 2 also shows that we do not change the linear classifier $U$, we only scale it by the gram matrix of concepts. This means our CBMs are *faithful* to the original classifier. By this, we obtain zero-shot CBMs that discover concepts and builds $W^{con}$,

both in a training-free manner, for any classifier, while also maintaining faithfulness to the original classifier and being flexible in the concept set.

### 4.2. Zero-Shot Decoding of Visual Features into Text

In this application, we aim to decode the visual feature vector $f$ for an image $I$, given by $f = F_v(I)$, into natural language sentence. This offers a text-based interpretation of what the visual features contain. We adapt the method introduced in (Tewel et al., 2021) for our purpose. Specifically, we first project the visual feature vector $f$ using the MLP to obtain $\tilde{f}$. That is, $\tilde{f} = \text{MLP}(f)$. Since $\tilde{f}$ is now in the same space as the text encoder $T$, we can measure its association to any encoded text. We utilize an off-the-self pretrained language decoder model (e.g., GPT-2), denoted as $G$, to generate open-ended text. We keep $G$ frozen to maintain its language generation capabilities and instead use prefix-tuning (Li & Liang, 2021) to guide $G$ to generate a text that maximizes the similarity with the transformed visual feature vector $\tilde{f}$. Specifically, we attach a set of learnable "virtual" tokens to $G$. Denote the generated output text of $G$ for one iteration as $h^j$, where $j$ represents the iteration number. $h^j$ is encoded with the text encoder $T$, to yield a vector $y^j$, That is, $y^j = T(h^j)$. As $y^j$ and $\tilde{f}$ are now in the same embedding space, we maximize the cosine similarity between them in order to update the learnable tokens. We perform this process for several iterations. An overview of this process is shown in Figure 4. As this process is not the core contribution of our work, we leave more details to Section E of the appendix, which also includes a detailed illustration in Figure 9.

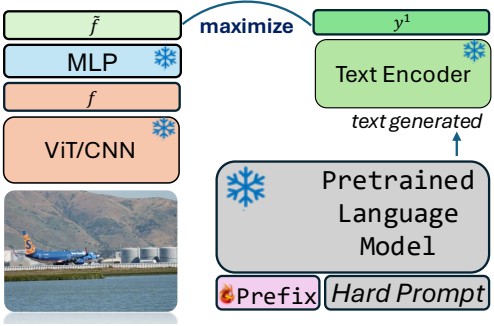

*Figure 4.* To decode visual features into natural language using any pretrained language decoder (e.g., GPT-2), we apply prefix tuning while keeping the language decoder fixed, generating text that maximizes the similarity with the visual features. The figure is shown for iteration $j = 1$

## 5. Experiments

We first provide results on our newly formulated visual classifiers. We use the ImageNet-1K dataset due to the

widespread availability of visual classifiers trained and evaluated on it. We apply our method on a diverse set of 40 visual classifiers. For CNNs, we consider the following family of models (each with several variants): Residual Networks (ResNets) (He et al., 2015), Wide ResNets (Zagoruyko & Komodakis, 2016), ResNeXts (Xie et al., 2016), ShuffleNetv2 (Ma et al., 2018), EfficientNetv2 (Tan & Le, 2021), Densely Connected Networks (DenseNets) (Huang et al., 2016), ConvNeXts (Liu et al., 2022) and ConvNeXtv2 (Woo et al., 2023). For Transformers, we consider the following family of models (each with several variants): Vision Transformers (ViTs) (Dosovitskiy et al., 2021), DINOv2 (Oquab et al., 2024), BeiT (Bao et al., 2022), the hybrid Convolution-Vision Transformer CvT (Wu et al., 2021), Swin Transformer (Liu et al., 2021b) and Swin Transformer v2 (Liu et al., 2021a). All models are pretrained on ImageNet-1K from the PyTorch[4] and HuggingFace (Wolf et al., 2020) libraries. Models with the subscript *pt* indicate that the model was pretrained on ImageNet-21k before being finetuned on ImageNet-1K. Models with a subscript *v2* are trained following the updated PyTorch training recipe (Vryniotis, 2021). Finally, BEiT, DINOv2 and ConvNeXtv2 are pretrained in a self-supervised manner before being finetuned on ImageNet-1k. Both the pretrained classifier and text encoder remain frozen, only the MLP is trained on the ImageNet training set following Eq. 1.

Performance is evaluated using the same protocol and dataset splits as the original classifier, specifically the 50,000 validation split of ImageNet-1K. For the text encoder, we use the MiniLM Sentence Encoder (Wang et al., 2020). Implementation details can be found in Section D of the appendix. Results are presented in Table 1. We report the replicated Top-1 accuracy of the re-formulated classifier with our method in the first column, the original Top-1 accuracy of the classifier in the second column, and the difference between them ($\Delta$) in the last column. As can be seen, the loss in performance as indicated by $\Delta$ is minimal, with an average drop in performance of approximately 0.2 points across all models. We refer to Appendix Section C for results on other models, and Appendix Section B for ablation studies on alternative text encoders.

**Zero-Shot Concept Bottleneck Models:** We report CBM evaluation results on the ImageNet validation set using the top-1 accuracy in Table 2. Our zero-shot CBMs (ZS-CBMs) outperforms all the supervised CBMs, even those based on the CLIP family, and sets a new state-of-the-art. Notably, even a simple ResNet-50 classifier trained solely on ImageNet already outperforms the CBM for the significantly more powerful ResNet-50 CLIP model trained on 400M samples. The best results are obtained by the ConvNeXtv2 and ViT-L/16 models, which achieve a top-1 accuracy of

---

[4]https://pytorch.org/vision/stable/models.html

| Model | Top-1 | Orig. | $\Delta$ |
|---|---|---|---|
| ResNet50 | 75.80 | 76.13 | $-0.33$ |
| ResNet50$_{v2}$ | 80.14 | 80.34 | $-0.20$ |
| ResNet101$_{v2}$ | 81.50 | 81.68 | $-0.18$ |
| ResNet101 | 77.19 | 77.37 | $-0.18$ |
| WideResnet50 | 78.35 | 78.47 | $-0.12$ |
| WideResNet50$_{v2}$ | 81.17 | 81.31 | $-0.14$ |
| WideResNet101$_{v2}$ | 82.21 | 82.34 | $-0.13$ |
| DenseNet161 | 77.04 | 77.14 | $-0.10$ |
| DenseNet169 | 75.46 | 75.60 | $-0.14$ |
| EfficientNetv2-S | 84.04 | 84.23 | $-0.19$ |
| EfficientNetv2-M | 84.95 | 85.11 | $-0.16$ |
| ShuffleNetv2$_{x2.0}$ | 75.83 | 76.23 | $-0.40$ |
| ConvNeXt-Small | 83.42 | 83.62 | $-0.20$ |
| ConvNeXt-Base | 83.88 | 84.06 | $-0.18$ |
| ConvNeXt-B$_{pt}$ | 85.27 | 85.52 | $-0.25$ |
| ResNeXt50-32x4d | 77.44 | 77.62 | $-0.18$ |
| ResNeXt50-32x4d$_{v2}$ | 80.79 | 80.88 | $-0.09$ |
| ResNeXt101-64x4d | 83.13 | 83.25 | $-0.12$ |
| ResNeXt101-32x8d | 79.10 | 79.31 | $-0.21$ |
| ViT-B/16 | 80.70 | 81.07 | $-0.37$ |
| ViT-B/16$_{v2}$ | 84.94 | 85.30 | $-0.36$ |
| ViT-L/32 | 76.72 | 76.97 | $-0.25$ |
| ViT-L/16 | 79.56 | 79.66 | $-0.10$ |
| Swin-Base | 83.22 | 83.58 | $-0.36$ |
| Swinv2-Base | 83.72 | 84.11 | $-0.39$ |
| BeiT-B/16 | 84.54 | 85.06 | $-0.52$ |
| BeiT-L/16 | 87.22 | 87.34 | $-0.12$ |
| DINOv2-B | 84.40 | 84.22 | $+0.18$ |
| ConvNeXtV2-B | 84.56 | 84.73 | $-0.17$ |
| ConvNeXtV2-B$_{pt}$ | 86.07 | 86.25 | $-0.18$ |
| ConvNeXtV2-B$_{pt}$@384 | 87.34 | 87.50 | $-0.16$ |

*Table 1.* Performance comparison of our re-formulated classifiers for several models. Top-1 indicates our results of the new formulation, and Orig. denotes the original Top-1 accuracy. $\Delta$ represents the difference between them $\Delta =$ Top-1 - Orig.

| Method | Model | Top-1 |
|---|---|---|
| **Supervised CBMs** | | |
| LF-CBM | CLIP ResNet50 | 67.5 |
| LF-CBM | CLIP ViT-B/16 | 75.4 |
| LaBo | CLIP ResNet50 | 68.9 |
| LaBo | CLIP ViT-B/16 | 78.9 |
| CDM | CLIP ResNet50 | 72.2 |
| CDM | CLIP ViT-B/16 | 79.3 |
| DCLIP | CLIP ViT-B/16 | 68.0 |
| DN-CBM | CLIP ResNet50 | 72.9 |
| DN-CBM | CLIP ViT-B/16 | 79.5 |
| **Zero-Shot CBMs (Ours)** | | |
| ZS-CBM | ResNet50 | 73.9 |
| ZS-CBM | ResNet50$_{v2}$ | 78.1 |
| ZS-CBM | ResNet101 | 75.3 |
| ZS-CBM | ResNet101$_{v2}$ | 79.9 |
| ZS-CBM | WideResNet50 | 76.9 |
| ZS-CBM | WideResNet50$_{v2}$ | 79.2 |
| ZS-CBM | WideResNet101$_{v2}$ | 81.0 |
| ZS-CBM | DenseNet121 | 69.9 |
| ZS-CBM | DenseNet161 | 75.2 |
| ZS-CBM | EfficientNetv2-S | 83.0 |
| ZS-CBM | EfficientNetv2-M | 83.9 |
| ZS-CBM | ConvNeXt-Small | 81.9 |
| ZS-CBM | ConvNeXt-Base | 82.8 |
| ZS-CBM | ViT-B/32 | 73.3 |
| ZS-CBM | ViT-B/16 | 79.3 |
| ZS-CBM | ViT-B/16$_{v2}$ | 83.2 |
| ZS-CBM | Swin-Base | 82.2 |
| ZS-CBM | Swinv2-Base | 82.6 |
| ZS-CBM | ViT-B/16$_{pt}$ | 81.5 |
| ZS-CBM | BeiT-B/16 | 83.0 |
| ZS-CBM | DINOv2-B | 82.6 |
| ZS-CBM | ConvNeXt-B$_{pt}$ | 84.0 |
| ZS-CBM | ConvNeXtV2-B$_{pt}$ | 84.9 |
| ZS-CBM | BeiT-L/16 | 86.2 |
| ZS-CBM | ConvNeXtV2-B$_{pt}$@384 | 86.3 |
| ZS-CBM | ViT-L/16$_{v2}$ | **86.3** |

*Table 2.* Performance of Zero-Shot CBMs (ZS-CBM) on ImageNet validation set for several classifiers.

86.3. All models show close to original accuracy, which means we can transform any classifier to be inherently interpretable without much loss in performance.

In Figure 5, we present qualitative examples of a selection from the top concepts responsible for the prediction, along with their weight importance. The weight importance is calculated by multiplying the concept activation with its corresponding weight to the predicted class. By observing the second example, we see that the image is predicted as a "goose" because it has duck-like features, it is in the size of a swan, and it has bird and pigeon-like features. Similarly, we can see that the third image is a "spotted salamander" due to its lizard-like structure, a head that looks like the frog's head, a snake-like tail and spots like a leopard. In the last image, we can see that the prediction is "ostrich" due to its

camel-like hump, mammal-like size, piegeon-like body, the horse-riding-like feature, and is a type of a bird. In Figure 6, we present a probability distribution of global class-wise concepts. These are concepts detected for all images of a specific class, along with their frequency. We consider two semantically similar classes but distinctively different: "hammerhead shark" and a "tiger shark". We highlight in yellow the top concepts in "hammerhead shark" that are not present in "tiger shark". These concepts are "harpoon" and "lobster hammer", both which are distinctive to the head of the hammerhead shark and drive its prediction.

**Zero-Shot Decoding of Visual Features:** We now evaluate the performance of the textually decoded visual features. Since the ground-truth content of visual features is unknown, we rely on COCO captions (Lin et al., 2014) for evalua-

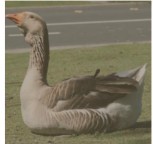
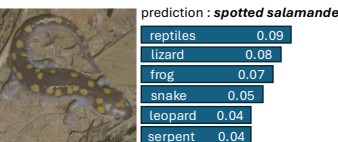
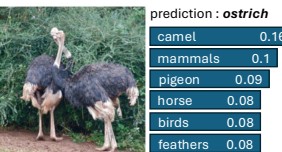

*Figure 5.* Qualitative examples of our zero-shot CBMs. We show the top-detected concepts, each with their corresponding importance score to the prediction. All examples use the 20K concept set, except for the first which uses the ImageNet LF-CBM concept set

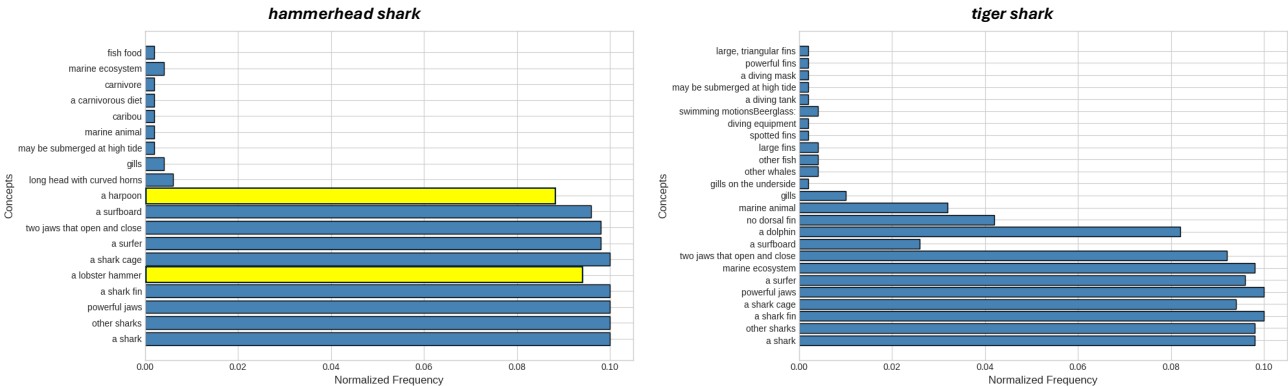

*Figure 6.* Global class-wise interpretability analysis with our Concept Bottleneck Model. We highlight in yellow the top concepts in "hammerhead shark" that are not present in "tiger shark", and therefore distinctive to "hammerhead shark".

tion. While these captions may not perfectly correspond to the actual information represented by a visual feature vector—potentially differing from what a human annotator expects—they provide a reasonable approximation for this purpose. Note that the COCO dataset differs in distribution than ImageNet, as a single image may contain many objects, interactions between them, and potentially categories not included in ImageNet (*e.g.,* person). Therefore, it also serves as a way to evaluate generalization of our method to other datasets, given that we only used the ImageNet class names for training. Since we do not train any model on the ground-truth image captions provided by COCO, we use zero-shot image captioning as a benchmark. We present results on the widely used "Karpathy test split" with various vision classifiers.

As baselines, we compare our approach against existing methods in zero-shot image captioning, specifically Zero-Cap (Tewel et al., 2021) and ConZIC (Zeng et al., 2023), both which use CLIP. For evaluation, we employ standard natural language generation (NLG) metrics: BLEU-4 (B@4) (Papineni et al., 2002), METEOR (M) (Banerjee & Lavie, 2005), ROUGE-L (R-L) (Lin, 2004), CIDEr (C) (Vedantam et al., 2014), and SPICE (S) (Anderson et al., 2016). Results are shown Table 3. ConVeXtv2 achieves state-of-the-art performance on CIDEr and SPICE, the two most critical metrics for evaluating image captioning systems. Even with a simple ResNet-50 vision encoder trained on ImageNet-1K (1.2 million images), our approach outperforms the baseline

methods on CIDEr and SPICE, despite the latter utilizing the significantly more powerful CLIP vision encoder, trained on 400 million image-text pairs. We also present qualitative examples of the decoded visual features for different models in Figure 7. We can see from the first example, that BeiT-L/16 contains both the vegetables and dog in its features, while ConvNexTv2 only focuses on the vegetables, and ViT-B/16 and DINOv2 only focus on on the dog. Although some generations lacks semantic correctness (e.g., who was born in a dog), they are still meaningful enough for humans to understand and reason. This application allows a user-friendly way of interpreting visual features, applicable even to a layman user.

Note that our results in Table 3 are outperformed by the baseline ZeroCap on the BLEU-4 (B4) and METEOR (M) metrics. However, it is important to note that B4 and M are n-gram overlap-based metrics. They assume that the generated caption follows a specific structure and style. We verify this by revisiting an old image captioning paradigm termed as *compositional captioning* (Kulkarni et al., 2011), and later revived with deep learning methods (Lu et al., 2018). In compositional captioning, a set of image-grounded concepts (such as attributes, objects and verbs) are first detected, and a language model is then used to compose them into a natural sounding sentence. With the current advancements of Large Language Models (LLMs), we can use an LLM as a compositioner. Specifically, we detect the top concepts and verbs to the image using the concept discovery method

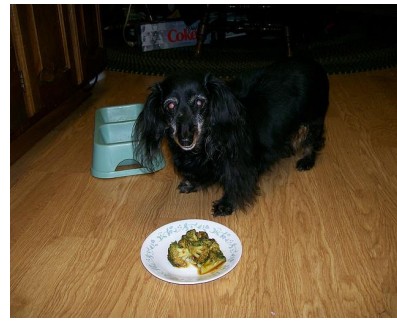

**BeiT-L/16:** broccoli and vegetables, the dog enjoying a puppy's favorite vegetable
**ConvNextv2:** broccoli caulifol, cabbage or spinach
**ViT-B/16:** Gordon, who was born in a dog and Labrador puppy
**DINOv2-B:** puppy british dog and toys

**BeiT-L/16:** shower curtain, bathing room and toilet.
**ConvNextv2:** shower curtain toilet seat and toilets, which were made by the bath.
**ViT-B/16:** shower curtain in the bathroom, and a small bathtub.
**DINOv2-B:** showers curtain seating in the toilets at showering.

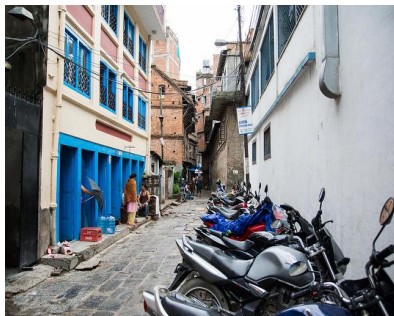

**BeiT-L/16:** motorcycles with the motorized scooters and electric motors
**ConvNextv2:** motorbike, scooter and motorcycle scrapping
**ViT-B/16:** motorcycles and motor scooters, motorcycle engines
**DINOv2-B:** motorcycles propelled by the propulsion motorless robotic scrappers

*Figure 7.* Qualitative examples of decoded visual features from different visual classifiers, into natural text.

| Model | B4 | M | R-L | C | S |
|---|---|---|---|---|---|
| ZeroCap | **2.6** | **11.5** | — | 14.6 | 5.5 |
| ConZIC | 1.3 | 11.5 | — | 12.8 | 5.2 |
| DenseNet161 | 1.50 | 10.2 | 20.4 | 15.8 | 6.3 |
| ResNet50 | 1.43 | 10.2 | 20.3 | 15.9 | 6.2 |
| WideResNet50 | 1.40 | 10.2 | 20.4 | 16.0 | 6.4 |
| WideResNet101$_{v2}$ | 1.50 | 10.4 | 20.5 | 16.6 | 6.4 |
| ResNet101$_{v2}$ | 1.48 | 10.4 | 20.6 | 16.7 | 6.5 |
| ResNet50$_{v2}$ | 1.47 | 10.5 | 20.6 | 16.8 | 6.5 |
| ConvNeXt-B$_{pt}$ | 1.50 | 10.6 | 20.8 | 17.2 | 6.7 |
| DINOv2-Base | 1.50 | 10.7 | 21.0 | 17.3 | 6.7 |
| ViT-B/16$_{v2}$ | 1.50 | 10.5 | 20.9 | 17.3 | 6.5 |
| BeiT-L/16 | 1.50 | 10.6 | 20.9 | 17.6 | 6.9 |
| ViT-B/16$_{pt}$ | 1.50 | 10.7 | 20.9 | 17.7 | 6.9 |
| ConvNeXtV2-B$_{pt}$@384 | 1.60 | 10.7 | **21.1** | **17.9** | **6.9** |

*Table 3.* Zero-Shot Image Captioning Performance

| Model | B4 | M | R-L | C | S |
|---|---|---|---|---|---|
| ZeroCap | 2.6 | 11.5 | — | 14.6 | 5.5 |
| ConZIC | 1.3 | 11.5 | — | 12.8 | 5.2 |
| **Ours** | | | | | |
| DenseNet161 | 4.20 | 12.5 | 30.1 | 17.0 | 6.6 |
| ResNet50 | 4.10 | 12.5 | 30.1 | 17.0 | 6.6 |
| WideResNet50$_{v2}$ | 4.20 | 12.7 | 30.3 | 17.7 | 6.9 |
| ResNet101v2 | 4.30 | 12.6 | 30.1 | 18.0 | 6.8 |
| ConvNeXt-Base | 4.50 | 12.7 | 30.3 | 18.1 | 6.9 |
| ResNet50$_{v2}$ | 4.50 | 12.8 | 30.5 | 18.4 | 6.9 |
| EfficientNetv2-S | 4.40 | 12.7 | 30.4 | 18.6 | 6.9 |
| ViT-B/16$_{pt}$ | 4.50 | 12.8 | 30.2 | 18.7 | **7.2** |
| ConvNeXtV2-B$_{pt}$@384 | 4.40 | 12.7 | 30.2 | 18.7 | **7.2** |
| BeiT-B/16 | 4.50 | 12.8 | 30.3 | **18.9** | 7.1 |
| DINOv2-Base | **4.60** | **13.0** | **30.7** | 18.7 | 7.1 |

*Table 4.* Composition Captioning Performance across models.

introduced in Section 4.1 and shown in Figure 3, and feed them, along with their similarity scores, to an LLM. For the concepts, we use the same concept set as in Section 4.1. We also add a list of the most common verbs[5] in English to the pool. This allows us to cover all possible words and interactions. We prompt the LLM to utilize the provided information to compose a sentence, given a very few in-context examples from the COCO captioning training set (in our experiments, we use 9 examples). This allows us to generate sentences adhering to a specific style and structure. For this experiment, we used GPT4o-mini (OpenAI, 2024) as our LLM, as it is fast and cost-efficient. In the prompt, we explicitly instruct the LLM to refrain from reasoning or generating content based on its own knowledge or assumptions, and that all its outputs must be strictly grounded in the provided concepts, verbs, and score importances. We provide details on the exact prompt we used in Appendix Section G. Results are shown in Table 4. While results on

CiDEr and SPICE are incremental compared to results in Table 3, the n-gram metrics (B4, M and R-L) are boosted, which verifies our hypothesis about the low scores of B4 and M in Table 3 compared to baseline methods.

## 6. Conclusion

In this work, we presented a method to transform visual classifiers such that they can be queried via open-set text queries. In two prominent applications, zero-shot CBMs and decoding of visual features into natural language, we achieve state-of-the-art results. Our method can be applied to various other applications that were previously restricted to CLIP models. Our work thus removes this restriction, being applicable to any visual classifier. Finally, as with any research work, this study has its own limitations which should be clearly acknowledged. We discuss limitations of our method in Appendix Section A.

[5]https://github.com/datmt/English-Words-Updated

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

# Appendix

## A. Limitations and Ethical Concerns

As with any research work, our study has its own set of limitations that should be transparent and acknowledged. In particular, we identify two primary limitations of our method: the first concerns wrong semantic associations of class names in CBMs, while the second pertains to the limited generalization of our method to fine-grained datasets.

We start by addressing the first limitation of wrong semantic associations of class names in CBMs. This issue is closely tied to the choice of concept set used. When using the 20K most common words in English as our concept set, we observe that class names get associated to either wrong semantically-related concepts, or with components of the class name itself, making the detected concepts in CBMs less meaningful. Figure 8 illustrates such cases. In the first example, the top-detected concepts for the prediction "african grey" are concepts that are direct components of the class name itself (grey, african), variations of these words (gray, africa), or incorrect semantic associations with the word "african" that do not pertain to the bird itself (Ethiopian, Tanzania). Similarly, in the second example, the bird drake is linked to artist-related concepts (Rihanna, Robbie, lyric). This occurs because the bird "drake" is less familiar to the text encoder than the artist "drake". In fact, a google search with the word "drake" yields directly the artist rather than the bird. In the third example, a similar issue arises with the animal cock, leading to associations with male reproductive terms. For each example, we also report the total logit score of the prediction.

However, it is worth noting the following:

1. Meaningful concepts such as "duck" (second example) and "hen" (third example) are still detected among the top concepts

2. The incorrect semantic associations contribute only a negligible portion of the total logit, accounting for approximately 0.01% of the overall prediction score.

3. This issue is considerably less severe when using alternative concept sets, such as the LF-CBM concept set tailored for ImageNet.

4. This issue also appears in CLIP-based CBMs and hence not unique to our approach

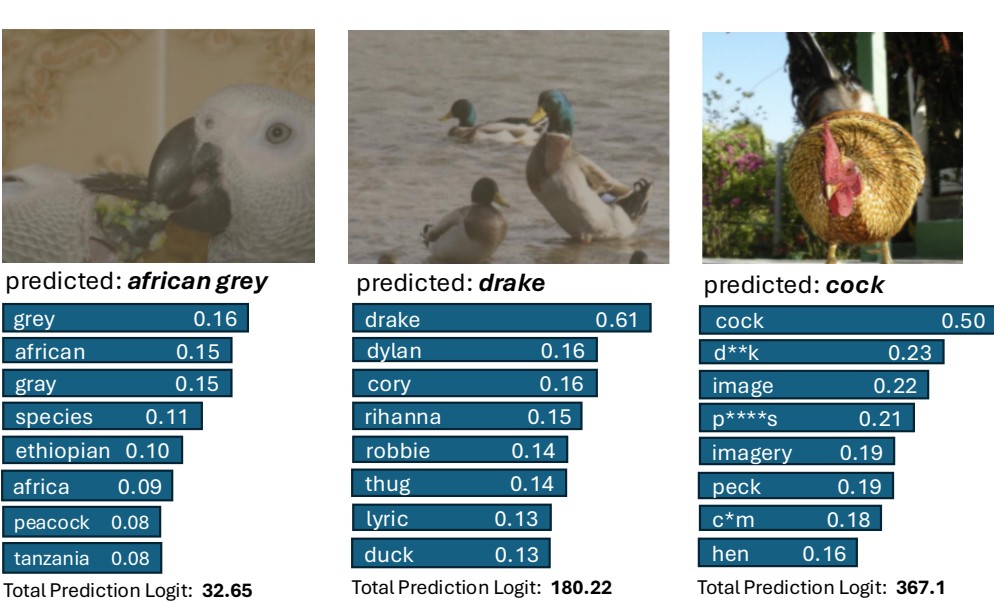

*Figure 8.* Limitations of our method is wrong semantic concept association in CBMs

The second limitation is the lack of generalization to fine-grained datasets. We test our ImageNet-trained reformulated classifiers to perform zero-shot transfer to the Places365 dataset (Zhou et al., 2017) for scene classification. It is important

to note that Places365 is known to induce low performance, even when using supervised training with the powerful CLIP model, making it a challenging classification dataset. To perform zero-shot transfer, we convert the class names of Places365 into text prompts of the form: "a scene of a {class} location" which are then encoded using the text encoder $T$ to obtain the classification weights $U_{places}$. The image is encoded with the respective vision encoder to obtain the visual features $f$, which are then projected via the MLP to yield $\tilde{f}$. Zero-shot classification is performed using $\text{argmax}(\tilde{f}.U_{places}^T)$. Results on the Places365 validation set are shown in Table 5. We report the supervised learning performance by training a linear probe on top of the CLIP model features. The supervised top-1 accuracy is 53.40 and 55.10 for ResNet50 and ViT-B/16 CLIP models, respectively. As demonstrated, these low top-1 accuracies suggest that Places365 is a particularly challenging dataset for classification. We then report the zero-shot performance of our method on several visual classifiers. The best-performing model achieves a top-1 zero-shot accuracy of 14.36%, much lower than the supervised performance. However, we observe that Transformer models based on large-scale pretraining perform better, a trend that aligns with findings in the out-of-distribution (OOD) literature. Moreover, our approach performs significantly better than a randomly shuffled classifier where we shuffle its weights across classes.

| Model | Top-1 (%) | Top-5 (%) |
|---|---|---|
| **Supervised** | | |
| CLIP RN50 Linear Probe | 53.40 | - |
| CLIP ViT-B Linear Probe | 55.10 | - |
| **Zero-Shot** | | |
| Shuffled Classifier | 0.24 | 1.54 |
| DenseNet161 | 10.55 | 25.18 |
| ResNet50 | 10.58 | 25.14 |
| ResNet101 | 10.97 | 25.42 |
| ViT-B/16 | 11.47 | 26.53 |
| ResNet50v2 | 11.66 | 27.40 |
| ConvNeXt-Base | 11.76 | 27.12 |
| Swin-Base | 12.12 | 26.83 |
| Swinv2-Base | 12.13 | 27.18 |
| EfficientNetv2-M | 12.31 | 28.33 |
| ViT-B/16v2 | 13.11 | 29.54 |
| DINOv2-Base | 13.96 | 32.35 |
| ConvNeXtv2 (pt) | 14.05 | 30.92 |
| BeiT-L/16 | 14.15 | 31.08 |
| ViT-B/16 (pt) | 14.22 | **32.18** |
| ConvNeXt (pt) | **14.36** | 31.64 |

*Table 5.* Top-1 and Top-5 zero-shot transfer results to Places365 for various models

## B. Ablation Studies on the Text Encoder

In this section, we present ablation studies using other text encoders. We test a variety of text encoders from the Sentence-BERT library (Reimers & Gurevych, 2019) using the ResNet50 visual classifier. Results are presented in Table 6. We observe that the choice of the text encoder has minimal effect on the performance, as even lower-performing text encoders are capable of understanding class names.

| Text Encoder | Top-1 (%) |
|---|---|
| DistilRoberta | 75.73 |
| MPNet-Base | 75.78 |
| MPNet-Base-MultiQA | 75.76 |
| MiniLM | **75.80** |

*Table 6.* Ablation studies on other text encoders

## C. Performance on Additional Models

We report performance on additional models that were not included in the main manuscript in Table 7.

| Model | Top-1 | Orig. | $\Delta$ |
|---|---|---|---|
| ConvNeXt-Tiny | 82.19 | 82.52 | $-0.33$ |
| ViT-B/32 | 75.40 | 75.91 | $-0.51$ |
| Swin-Small | 82.63 | 83.20 | $-0.57$ |
| Swinv2-Tiny | 81.44 | 82.07 | $-0.63$ |
| CvT-21 | 80.45 | 81.27 | $-0.82$ |
| ViT-L/16$_{\text{v2}}$ | 87.61 | 88.06 | $-0.45$ |
| Swinv2-Small | 83.32 | 83.71 | $-0.39$ |
| ViT-B/16$_{\text{pt}}$ | 83.55 | 84.37 | $-0.82$ |

*Table 7.* Performance of our reformulated classifiers for additional models

## D. Implementation Details

For the text encoder, we use the all-MiniLM-L12-v1[6] model available on the Sentence Transformers library (Reimers & Gurevych, 2019). This text encoder was trained on a large and diverse dataset of over 1 billion training text pairs. It contains a dimensionality of $m = 384$ and has a maximum sequence length of 256.

Our MLP projector is composed of 3 layers, the first projects the visual feature dimensions $n$ to $n \times 2$ and is followed by a Layer Normalization (Ba et al., 2016), a GELU activation function (Hendrycks & Gimpel, 2016) and Dropout (Srivastava et al., 2014) with a drop probability of 0.5. The second layer projects the $n \times 2$ dimensions to $n \times 2$ and is followed by a Layer Normalization and a GELU activation function. The final linear layer projects the $n \times 2$ dimensions to $m$ (the dimensions of the text encoder). We train the MLP projector with a batch size of 256 using the ADAM optimizer (Kingma & Ba, 2015) with a learning rate of 1e-4 that decays using a cosine schedule (Loshchilov & Hutter, 2017) over the total number of epochs. We follow the original image sizes that the classifier was trained on.

For the training images, we apply the standard image transformations that all classifiers were trained on which include a Random Resized Crop and a Random Horizontal Flip. For the validation images, we follow exactly the transformations that the classifier was evaluated on, which include resizing the image followed by a Center Crop to the image size that the classifier expects. Each model is trained on a single NVIDIA GeForce RTX 2080 Ti GPU.

## E. Process of Decoding Visual Features

We remind readers of the mapping function, denoted as MLP, that transforms the visual features $f$ into the same space as textual features, producing $\tilde{f}$. A pre-trained language model $G$ is then optimized to generate a sentence that closely aligns with $\tilde{f}$. To preserve the generative power of $G$, we keep it frozen and apply prefix-tuning (Li & Liang, 2021), which prepends learnable tokens in the embedding space. During inference, these tokens are optimized separately for each input. Our method builds upon the work of (Tewel et al., 2021).

A high-level overview of this process is illustrated in Figure 9. Using a pre-trained language model $G$, we prepend randomly initialized learnable tokens, referred to as prefixes, which guide $G$ to produce text that maximizes alignment with visual features. These learnable prefixes function as key-value pairs in each attention block, ensuring that every generated word can attend to them.

For each iteration $j$, at a timestep $ts$, we sample the top-$Q$ tokens from the output distribution of $G$, denoted as $G_{out}$, which serve as possible continuations for the sentence. These $Q$ candidate sentences are then encoded by a text encoder $T$, mapping them into the same embedding space as $\tilde{f}$. We compute the cosine similarity between each encoded sentence and $\tilde{f}$, resulting in $Q$ similarity scores. These scores are normalized with softmax and define a target distribution used to train $G_{out}$ via Cross-Entropy loss. The learnable prefixes are updated through backpropagation.

With the updated prefixes, $G$ is run again, and the most probable token is selected as the next word. This process is repeated

---

[6]https://huggingface.co/sentence-transformers/all-MiniLM-L12-v1

for a predefined number of timesteps (up to the desired sentence length) or until the $<.>$ token is generated. At the end of each iteration, a full sentence is generated. We conduct this process for 20 iterations, generating 20 sentences in total. The final output is chosen as the sentence with the highest similarity to the visual features $\tilde{f}$.

We also add the fluency loss from (Tewel et al., 2021) as well as other token processing operations. We refer readers to (Tewel et al., 2021) for more information. We use the small GPT-2 of 124M parameters as $G$. We also noticed that using a bigger $G$ (e.g., GPT-2 medium) does not enhance performance, indicating that a decoder with basic language generation knowledge is sufficient.

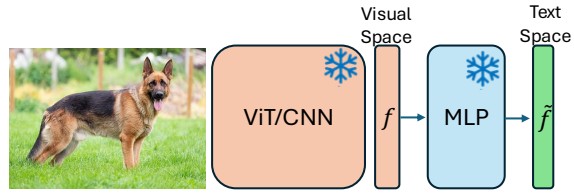

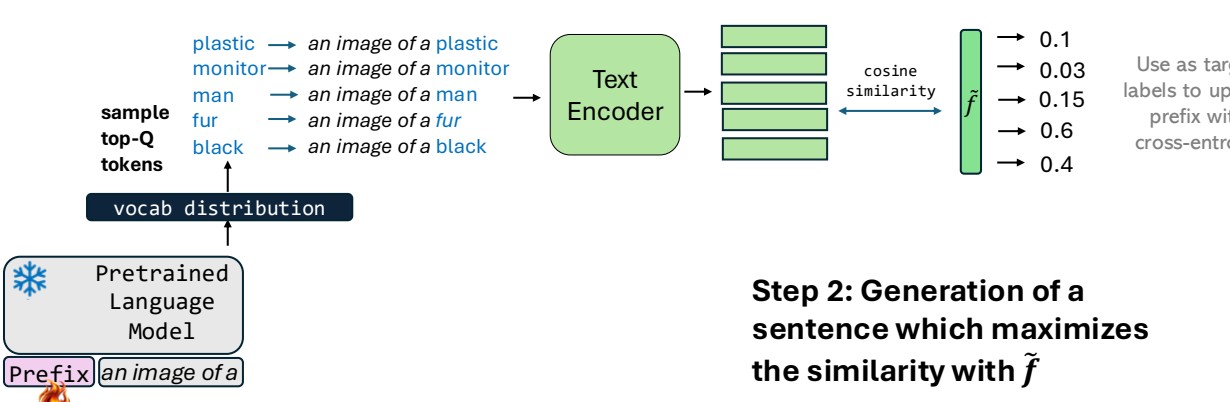

Figure 9. The process used to decode visual features of an image. The process is shown for the first timestep $ts = 1$ with a hard prompt set as "an image of a".

## F. Using only class names

We remind readers from Section 3 that we only use the class names to formate the text prompt for the text encoder when training the MLP. In practice, we can go beyond class names by using resources like a class hierarchy from WordNet (Lin, 1998) (the original source where ImageNet was extracted from), or class descriptions extracted from a LLM as in CuPL (Pratt et al., 2023) or VCVD(Menon & Vondrick, 2023). However, this approach would be considered as "cheating," as it fails to faithfully replicate the classifier. The original classifier implicitly learns the semantics, hierarchies, relationships and distinctive features of different classes. Explicitly providing additional information would not replicate the classifier faithfully, and would also force the classifier to focus on predefined features or those we intend it to learn. Moreover, this would also leak information to downstream tasks such as CBMs and textual decoding of visual features, compromising the fairness of evaluation. For instance, if class descriptions were used in the training, the concepts in CBMs would align with those specified in the training prompts. For these reasons, we refrain from using any other additional information than the class names. We use the class names provided from `https://gist.github.com/yrevar/942d3a0ac09ec9e5eb3a`.

## G. Compositional Captioning

The prompt we used for compositional captioning is as follows:

```
I will give you several attributes and verbs that are included in an image,
```

```
each with a score.  The score reflects how important (or how grounded) the
attrribute/verb is to the image, and higher means more important and grounded.
Your job is to formulate a caption that describes the images by looking at the
attributes/verbs with their associated scores.  You should not reason or generate
anything that is based on your own knowledge or guess.  Everything you say has
to be grounded in the attrbiutes/verbs and score importances.  Please use the
following the structure, style, and pattern of the following examples.  Example
1:  A woman wearing a net on her head cutting a cake.  Example 2:  A child
holding a flowered umbrella and petting a yak.  Example 3:  A young boy standing
in front of a computer keyboard.  Example 4:  a boy wearing headphones using
one computer in a long row of computers.  Example 5:  A kitchen with a stove,
microwave and refrigerator.  Example 6:  A chef carrying a large pan inside of a
kitchen.  Here are the attributes and scores:  {detected concepts with scores},
and these are the verbs and scores:  {detected verbs with scores}.
```

Next, we explore alternative concept sets in Compositional Captioning. In the main manuscript, we reported results using the 20,000 most common English words as our concept set. Since the LLM remains fixed and functions as a composer, integrating detected concepts and verbs grounded in the image into a caption, we can seamlessly substitute the concept set with any domain-specific concept set alternative. This allows for the generation of captions (here, decoded visual features) tailored to a specific domain. Here, we maintain the same set of verbs but explore the use of concepts specific to the ImageNet dataset. Since ImageNet lacks dedicated captions, we evaluate the domain-specific captioning by anticipating a decline in performance on the COCO captioning dataset. We use the ImageNet-specific concept set from (Oikarinen et al., 2023) and report zero-shot captioning performance in Table 8. As shown, we observe a decrease in all metrics. This shows that our method can readily decode visual features into text for any domain. Finally, also note that we can control the style of the generations by simply prompting the LLM to compose the concepts and verbs in a specific style (e.g., humorous, positive, negative).

| Method | B4 | M | R-L | C | S |
|---|---|---|---|---|---|
| ZeroCap | 2.6 | 11.5 | — | **14.6** | 5.5 |
| ConZIC | 1.3 | 11.5 | — | 12.8 | 5.2 |
| **Ours** | | | | | |
| MobileNetv3-L | 3.50 | 12.7 | 29.1 | 11.4 | 6.1 |
| ResNet50 | 3.60 | 12.7 | 29.3 | 12.0 | 6.0 |
| ResNet101v2 | 3.50 | 12.9 | 29.3 | 12.2 | 6.3 |
| WideResNet101v2 | 3.70 | 12.9 | 29.6 | 12.4 | 6.2 |
| ConvNeXt-Base | 3.80 | 12.8 | 29.5 | 12.7 | 6.2 |
| EfficientNetv2-S | 3.70 | 12.9 | 29.6 | 12.9 | 6.3 |
| ViT-B/16 (pt) | 3.80 | 13.1 | 29.5 | 13.2 | 6.5 |
| BeiT-L/16 | **3.90** | **13.2** | **29.6** | 13.4 | **6.6** |

*Table 8.* Composition Captioning Performance using the ImageNet-specific LF-CBM concept set

