# OpenReview forum: "Transforming Visual Classifiers for Zero-Shot Text-Based Interpretability"
_ICML.cc/2025/Conference — Submitted to ICML 2025_

### Official Review · Reviewer_HxVm · 2025-02-17

**Overall Recommendation:** 3

**Summary:**

The authors introduce a method for increasing the interpretability of image classifiers by combining the image classifier with a text tokenizer and their own trainable FFN. Essentially, the image feature vector obtained from passing the image to the vision classifier is converted into the text embedding space by the proposed FFN. This new image/text encoded vector is compared against each of the encoded dataset classes using cosine similarity, producing a new distribution over the output classes. The FFN is trained using cross-entropy loss between the predicted class distribution and the aforementioned cosine similarity vector.

The two main contributions are that this allows for an increase in model interpretability without (1) needing additional labeled training data and (2) being specific to any particular vision encoder and text tokenizer.

The authors provide a number of experiments. (1) They show an average loss in accuracy of only about 0.2% when compared against the vanilla model their method is augmenting. (2). They compare against other methods they discuss in the paper. (3) They adapt their method for zero-shot captioning and provide an experiment. (4) They also provide additional experiments in the appendix.


## Update from rebuttal ##
During the discussion phase, I raised a question regarding the faithfulness highlighted by one of the other reviewers. It would have been ideal if the authors would have used an example with multiple classes instead of a binary class problem when responding to this question. The use of a binary class problem limits the insights gained from the experiment. The use of a two class problem essentially makes the concepts and the classes equivalent? Therefore, I lowered my score to a 3.

**Claims And Evidence:**

Yes

**Essential References Not Discussed:**

The appropriate references are included.

**Experimental Designs Or Analyses:**

The experimental setup is intuitive

**Methods And Evaluation Criteria:**

The paper is rigorously evaluated using many models.

**Other Comments Or Suggestions:**

n/a

**Other Strengths And Weaknesses:**

Strengths:
1) I found the use of the original image output distribution as the ground truth for the loss function to be a clever way to get around having to use labeled data and become specific to the architectures in use.
 2) All of the writing is very clear and easy to follow
 3) The figures are particularly useful
 4) I appreciate the authors going into detail as to why their approach did not perform as well on B@4 and Meteor metrics on the Zero-Shot evaluation.

This is a very solid submission that is organized well and presented clearly.

**Questions For Authors:**

During the discussion phase, I raised a question regarding the faithfulness highlighted by one of the other reviewers. It would have been ideal if the authors would have used an example with multiple classes instead of a binary class problem, which limits the insights gained from the experiment. The use of a two class problem essentially makes the concepts and the classes equivalent?

**Relation To Broader Scientific Literature:**

Much research is going into model interpretability, as neural networks are considered black boxes until some algorithm is introduced that allows the user to understand how the network came to a decision. There are many ways to approach model interpretability. This paper approaches it by examining how the features of an image, generated by a visual encoder (CNN, transformer), can be explained through text. This particular line of research has been explored previously. The key contributions of this paper is that the authors are presenting a way to provide textual interpretations of visual features without (1) needing labeled data and (2) being language model architecture dependent.

**Theoretical Claims:**

The main theoretical claim they make is with the loss function for the FFN. While I have not checked it myself, the presented equation is intuitive and makes sense in the context.

---

> ### Author Rebuttal · Authors · 2025-03-31
>
> Thank you very much for your time and effort in reviewing our paper, and for the thoughtful review and strong accept decision. We are delighted that you found our manuscript interesting and appreciated the importance and unique aspects of our work, such as the innovative, novel, and clever solution, the rigorous evaluations, clear writings, and detailed analysis. Your acknowledgment of the importance and strengths of our work is truly encouraging, and we are grateful for your supportive comments.

---

> > ### Comment · Reviewer_HxVm · 2025-04-06
> >
> > I think the idea of mapping the output distribution of the labels of any classifier into a text space is an important contribution. This was the main reason for my initial score of the paper.
> >
> > Several of the other reviewers have raised concerns regarding the faithfulness of the explanations. The paper is mainly evaluated using end-to-end results on the quantitative side. Would it be possible to perform a quantitative evaluation to assess if the extracted concepts are faithful. For example, reviewer bjB7 mentioned the issue below. However, I understand if it is not possible to generate a full set of new results before the end of the discussion period.
> >
> > 1. Quantitative experimental evaluation of the reliability and interpretability of the explanation is insufficient. The paper evaluates the proposed method mainly in terms of task accuracy. However, these evaluations do not guarantee the validity of the generated explanations. A fair comparison with existing methods should be made, e.g., by evaluating the quality of concepts and interventions as done in [c] and [d].

---

> > > ### Author Response · Authors · 2025-04-08
> > >
> > > We thank you for your comment. We address your concern below with experiments.
> > >
> > > First of all, previous works [R1] conduct the intervention experiments using the Waterbirds-100 dataset. This is a binary classification dataset which includes two classes: waterbirds and landbirds. The training images of waterbirds are on water backgrounds, and the training images of landbirds are on land backgrounds. However, the validation images do not have that correlation (the validation images of waterbirds are on land backgrounds and the validation images of landbirds are on water backgrounds). The model is assumed to learn the water-land correlation for classification on this task. By building a CBM, we can correct this bias by intervening in concepts in the CB layer. However, there are two problems with using the Waterbirds dataset in our work:
> > >
> > > 1) We cannot transform a classifier trained on this dataset with our method, because it is not possible to learn meaningful text representations (our MLP) using only two text labels (waterbird, landbird).
> > > 2) We also cannot test our ImageNet-trained models on this dataset because Waterbirds is a highly artificial dataset, and many of the samples create severe OOD samples for ImageNet trained classifiers.
> > >
> > > Therefore, we conduct the following experiment.
> > >
> > > We manually created our own dataset of waterbirds/landbirds from ImageNet validation images. Specifically, for the waterbird images, we consider classes of birds from ImageNet that are at least 90% of the time found in water backgrounds (we manually inspected 100 random training images of those birds to verify this). For those birds, we then select their images in the ImageNet validation set, that are found on land backgrounds. We perform a similar procedure for the landbird class.
> > >
> > > For the landbird class, we encountered an issue. While we found many classes of birds from ImageNet to have land backgrounds in their training images (almost all the time), we could not find many validation images of those birds to have water backgrounds (e.g., we could not find any image of the bird *robin* with water background in the ImageNet validation set). In order to solve this issue, we utilized a text-to-image generative model (Stable Diffusion 2.1) to generate images of those birds on water backgrounds. We ensured that the generated images of those birds have the correct physical and distinctive features of the bird.
> > >
> > > For all images, we always ensure that the background (water/land) is clearly visible. This leads us to a validation dataset of 140 images (70 images for each class):
> > >
> > > - Waterbirds: 70 images from ImageNet validation set
> > > - Landbirds: 15 images from ImageNet validation set, and 55 generated from Stable Diffusion.
> > >
> > > We create our ZS-CBM using the two class labels: "an image of a waterbird" (for the waterbird class) and "an image of a landbird" (for the landbird class), using the same concept set from [R1]. The ZS-CBM achieves a low accuracy as shown in the Table below, which indicates the bias that the model uses. To correct this, we intervene in the concepts in the CB layer, following the setup from [R1]:
> > >
> > > - Intervention R: We zero-out activations of any bird concepts from the bottleneck layer, and expect the accuracy to drop.
> > > - Intervention K: We keep activations of bird concepts as they are, but scale down the activations of all remaining concepts (we multiply them by 0.1), and expect the accuracy to increase.
> > >
> > > The results are presented below for some models and show the success of our intervention experiments:
> > >
> > > |Model| Original CBM | Intervened (R)↓ | Intervened (K) ↑ |
> > > |-|-|-|-|
> > > |BeiT-B/16:| 54.29| 41.43 **(-12.86)**| 58.57 **(+4.28)** |
> > > |BeiT-L/16| 52.86| 44.29 **(-8.57)**| 58.57 **(+5.71)**|
> > > |ConvNextv2_pt@384| 53.57| 42.14 **(-11.43)**| 59.29 **(+5.72)**|
> > > |ConvNext_Base_pt |53.57| 42.86 **(-10.71)**| 58.57 **(+5.0)**|
> > > |DiNOv2|52.86|43.57 **(-9.29)**|59.29 **(+6.43)**|
> > >
> > >
> > > [R1] Discover-then-Name: Task-Agnostic Concept Bottlenecks via Automated Concept Discovery

---

### Official Review · Reviewer_bjB7 · 2025-03-11

**Overall Recommendation:** 1

**Summary:**

The paper aims to make arbitrary image classifiers interpretable by textual explanations. The paper points out as a challenge that existing methods rely on CLIP, which may limit applications. To this end, the paper maps image features of arbitrary visual classifiers to text features of off-the-shelf language models by transforming the image features with MLP and training the relationship between the visual features and the class labels in the text space to be similar to the original classification result. The paper proposes a zero-shot concept bottleneck model and zero-shot text decoding as applications of this mapping. Experiments quantitatively confirm that the proposed method outperforms baselines, including CLIP-based methods, in terms of accuracy and qualitatively evaluate the output text-based explanations.

## Updates after rebuttal
I would like to maintain my rating according to the evaluation below:

As discussed in the Claim and Evidence section of my initial review, my primary concern is the faithfulness of the explanations provided by the proposed method. In summary, after rebuttal, there are currently three critical problems in faithfulness.

- Since the proposed method aligns visual and text features at the distribution level via MLP, the guarantee of the semantic correspondence between the proposed explanation and the visual feature is weak in theory.
- Quantitative experimental evaluation of the explanation's reliability and interpretability is insufficient. Task accuracy is not appropriate for evaluating explanations.
- The proposed method shifts the explanation dependency from CLIP to another language model, and the essential difference from the approach that uses CLIP is not clear because there is no empirical comparison in this regard.

In addition to them, it became clear in the discussion that there is a lack of verification on fine-grained datasets. The authors present the results of the Places dataset in the zero-shot transfer setting of the ImageNet pre-trained model in Appendix A, but they avoid evaluating the model trained on Places. If the authors' claim is correct, the proposed method should also work effectively on such fine-grained datasets. This evaluation is quite important because the main motivation of this work is to make the custom visual models explainable, and such custom models are often used for specific (e.g., fine-grained) tasks.

Based on these discussions, I believe the current rating should be maintained.

**Claims And Evidence:**

- **Claim 1. Parameter efficiency**.
  - The paper claims that the proposed method is efficient because it optimizes only MLP.
  - However, there is no basis for this claim because existing methods (e.g., Text-to-Concept [a]) have the same or lower number of parameters.
- **Claim 2. Label-free training**
  - The paper claims that the proposed method does not require any label annotations when training MLPs.
  - This claim is valid because the algorithm in Section 3 trains MLPs for feature projection in a manner similar to self-learning without using labels.
- **Claim 3. Faithfulness of explanation**
  - The paper claims that projecting image features of any model onto text features can provide a reliable explanation.
  - This claim has not been adequately evaluated. The reasons are as follows:
    - The proposed method learns a mapping between the image feature space and the text feature space using the text features of the class labels, but the basis for the validity of this mapping is insufficient. That is, since the image features and text features are not learned in a one-to-one correspondence, the paper cannot guarantee that the image features projected by MLP correspond semantically exactly to text features. For example, [b] guarantees this with cycle consistency by inverse mapping (i.e., text-to-image) over the feature spaces.
    - Quantitative experimental evaluation of the reliability and interpretability of the explanation is insufficient. The paper evaluates the proposed method mainly in terms of task accuracy. However, these evaluations do not guarantee the validity of the generated explanations. A fair comparison with existing methods should be made, e.g., by evaluating the quality of concepts and interventions as done in [c] and [d].
    - The paper points out in L080 (right) that existing CLIP-based methods are biased against CLIP but lack discussion and rationale for this negative impact and the advantages of the proposed method. The proposed method uses an external language model instead of CLIP. In other words, the feature transformation by MLP is affected by the bias of the external language model. In fact, Table 1 shows that the use of textual features degrades the performance on average, indicating that the transformation is not perfect and that some changes have occurred. Without discussing the impact of these changes on interpretability, it is difficult to claim that the explanation is faithful.
- **Claim 4. Architecture Independency**
  - The paper claims that the proposed method is applicable to arbitrary models.
  - Section 5 indeed provides experimental results in terms of performance, but there is insufficient evidence to show that it can provide a reasonable explanation for interpretability for each architecture.

### Reference
- [a] Moayeri, Mazda, et al. "Text-to-concept (and back) via cross-model alignment." ICML 2023.
- [b] Kim, Siwon, et al. "Grounding counterfactual explanation of image classifiers to textual concept space." CVPR 2023.
- [c] Koh, Pang Wei, et al. "Concept bottleneck models." ICML 2020.
- [d] Yang, Yue, et al. "Language in a bottle: Language model guided concept bottlenecks for interpretable image classification." CVPR 2023.

**Essential References Not Discussed:**

There is a lack of discussion with existing research on making any image classifier explainable. I recommend to include the following literature in the discussion.

- Kim, Siwon, et al. "Grounding counterfactual explanation of image classifiers to textual concept space." CVPR 2023.
- Laguna, Sonia, et al. "Beyond concept bottleneck models: How to make black boxes intervenable?." NeurIPS 2024.
- Yang, Xingyi, and Xinchao Wang. "Language Model as Visual Explainer." NeurIPS 2024.
- Balasubramanian, Sriram, Samyadeep Basu, and Soheil Feizi. "Decomposing and interpreting image representations via text in vits beyond CLIP." NeurIPS 2024.
- Tan, Andong, Fengtao Zhou, and Hao Chen. "Explain via any concept: Concept bottleneck model with open vocabulary concepts." ECCV 2024.

**Experimental Designs Or Analyses:**

The validity of the explanations provided by the proposed method has not been experimentally evaluated. For example, examining the intervention on the concept and the rate of recovery of the true concept would be helpful in evaluating the explanations. Also, since the dataset is limited to ImageNet and COCO, it has not been verified that the method generalizes to a variety of datasets and domains.

**Methods And Evaluation Criteria:**

The proposed method is implemented by a simple MLP and cosine similarity. The paper quantitatively evaluates the proposed method in terms of task accuracy and qualitatively evaluates the explanations. On the other hand, since the explanations are not evaluated quantitatively, the claims regarding the quality and faithfulness of the explanations are not verified.

**Other Comments Or Suggestions:**

Nothing to report.

**Other Strengths And Weaknesses:**

Nothing to report.

**Questions For Authors:**

See the section "Claims and Evidence" and address the concerns.

**Relation To Broader Scientific Literature:**

The goal of this paper is to obtain interpretability by connecting image and language models. In this regard, the idea of mapping image features to text features with MLP or linear layers has been reported in several existing studies [a,h]. The main novelty of this paper is to show that a textual explanation can be provided by mapping image features to text features without using CLIP. However, the contribution is limited due to the lack of evidence for the validity of the provided explanation, as discussed above.

### Reference
- [h] Merullo, Jack, et al. "Linearly mapping from image to text space." ICLR 2023.

**Theoretical Claims:**

There is no theoretical claim in this paper. The paper claims that the trained MLP can transform image features into text features, but this has no theoretical guarantee.

---

> ### Author Rebuttal · Authors · 2025-03-30
>
> We thank the reviewer for his time in reviewing our paper. We clarify your concerns below.
>
> > Existing methods (e.g., Text-to-Concept [a]) have the same or lower number of parameters.
>
> As mentioned in L70-L84 in the related work, while the parameters may be the same or less, Text-to-Concept is not faithful (largely biased towards CLIP), and not label-free (requires supervision from CLIP features as the ground-truth). Efficiency is not the sole objective of our work.
>
> > the idea of mapping image features to text features with MLP or linear layers has been reported in several existing studies [a,h]
>
> Work [a] and its drawbacks are discussed clearly in the related work. We also mention how our method is different and how it tackles the problems of [a]. We refer you to L71-L84 (right). [h] is a work that uses image annotated captions as the training objective and therefore is very similar to DeVIL and the other mentioned works in Section 2 and suffers from the same limitations. We also mention how our method is different and how it tackles their problems. We refer you to L106 (left)-L61. The issues that you have raised are inherent in the literature and are thoroughly discussed in the related work. Our method is explicitly made to tackle the problems that these methods have.
>
> > The paper points out in L080 (right) that existing CLIP-based methods are biased against CLIP but lack discussion and rationale for this negative impact and the advantages of the proposed method.
>
> We discuss this thoroughly in L21-29 (right).
>
> > Table 1 shows that the use of textual features degrades the performance on average, indicating that the transformation is not perfect and that some changes have occurred.
>
> The performance drop is negligible (0.2 points drop on average across all models). In our view, this negligible drop does not degrade performance. Furthermore, the negligible loss is compensated by unlocking text-based interpretations to vision models, all in a zero-shot manner.
>
> > Since the image features and text features are not learned in a one-to-one correspondence, the paper cannot guarantee that the image features projected by MLP correspond semantically exactly to text features.
>
> We explicitly align the distributions rather than the features. The loss we use guarantees that the distributions are aligned. Using a one-to-one feature-to-feature mapping as the reviewer suggests will ignore the relation of other classes to the image, and will ignore the classifier’s reasoning process. As a very simple example, consider an image of a dog, with a chair present in the background. The softmax distribution for this sample would assign a high probability to the dog class and a moderately high, yet non-negligible probability to the chair class. Therefore, using distribution alignment loss considers the presence of both these classes. It cares about the distribution of the image over all classes, which is also how the classifier reasons and makes decisions. That is why, using distribution alignment loss rather than feature alignment loss is a faithful, effective and valid way for our approach.
>
> > It has not been verified that the method generalizes to a variety of datasets and domains.
>
> We did verify the zero-shot generalization of our MLPs trained solely on ImageNet class names, to the COCO dataset. We mention this in detail in L360-L366, where we describe how the two datasets differ in distribution. Furthermore, we showed zero-shot classification generalization experiments to other datasets (Places365) in the supplementary material (Table 5). We were also transparent and acknowledged in our manuscript that the classification generalization of our method to more fine-grained datasets is a limitation of our method.
>
> > Examining the intervention on the concept and the rate of recovery of the true concept would be helpful in evaluating the explanations. The claims regarding the quality and faithfulness of the explanations are not verified.
>
> We thank the reviewer for the valuable comment. We agree that evaluating the intervention on concepts could offer additional insights into the interpretability and reliability of the explanations. However, evaluating with interventions on ImageNet poses challenges due to the lack of intervention annotations (the expected biases in ImageNet classes). We therefore chose the core, standard way of evaluating CBMs which is task accuracy. For decoding of visual features, the main issue is, that we don't know what the model uses for reasoning (no ground truth). There is currently a gap in the literature regarding a proper benchmark for such models. We have discussed the limitation of this evaluation in L355 (left) - L359 in the main manuscript. As there is no ground-truth of what the model uses for reasoning, using annotated captions is the best approximation we can have.

---

> > ### Comment · Reviewer_bjB7 · 2025-04-03
> >
> > Thank you for the response.
> >
> > > As mentioned in L70-L84 in the related work, while the parameters may be the same or less, Text-to-Concept is not faithful, ...  and not label-free ... Efficiency is not the sole objective of our work.
> >
> > If there is no baseline to compare and no technical novelty, then this is simply the result of a naive implementation, and efficiency is hardly the main contribution of the paper. I would recommend introducing efficiency as a secondary benefit and lowering the tone in writing.
> >
> > > Work [a] and its drawbacks are discussed clearly in the related work. We also mention how our method is different and how it tackles the problems of [a]. We refer you to L71-L84 (right). ...
> >
> > I've already read the discussion and understood the difference between methods. Here, I've discussed the main technical novelty of this paper in the context of broader scientific literature, not to mention the lack of discussion. In this sense, the main technical novelty of this paper is to show that a textual explanation can be provided by mapping image features to text features without using CLIP. However, as mentioned in the initial review, this technical novelty has not been validated sufficiently.
> >
> > > We discuss this thoroughly in L21-29 (right).
> >
> > Of course, I've already read the mentioned discussion. However, this discussion does not seem to address my concerns about the following: "Since the linear layer maps feature to the CLIP space, T2C is strongly biased towards interpreting the CLIP model rather than the original classifier." In other words, what is the difference between the explanation given by CLIP and that given by the proposed method? And why is the proposed method "faithful" even though it also performs feature transformations in MLP? The current paper seems to provide little convincing evidence in this regard, as “faithfulness” is not defined or adequately evaluated.
> >
> > > The performance drop is negligible (0.2 points drop on average across all models).
> >
> > Sorry for the lack of clarity. Here, I discussed about the faithfulness of the explanation, not about the negligibility of the performance drop. If there is some changes in performance, there is some changes in the feature. Under such situation, how does the proposed method guarantee the correctness of the explanation? Is the explanation faithful?
> >
> > > That is why, using distribution alignment loss rather than feature alignment loss is a faithful, effective and valid way for our approach.
> >
> > Thank you for the clarification. However, faithfulness has not been proved theoretically and experimentally. Also, the example you presented assumes a one-to-one correspondence between visual class objects and text, but text data is flexible enough to represent whole image including the relationship between classes. My question is, why can the proposed method guarantee that a feature converted uni-directionally from image to text is faithful?
> >
> > > Furthermore, we showed zero-shot classification generalization experiments to other datasets (Places365) in the supplementary material (Table 5). We were also transparent and acknowledged in our manuscript that the classification generalization of our method to more fine-grained datasets is a limitation of our method.
> >
> > Thanks for the additional explanation. The limited generalization in fine-grained recognition is critical because fine-grained recognition is considered as the main application of custom image recognition models in industry, which is the target of this paper as presented in Introduction. This explanation reveals that there is a gap between the motivation of the paper and the results of the proposed method.
> >
> > > We therefore chose the core, standard way of evaluating CBMs which is task accuracy.
> >
> > Given that Black-box models achieve good accuracy, task accuracy cannot be used as a proxy measure of explainability. For example, [e] reports that models that explain the prediction via text sentences can achieve high accuracy even when the textual explanation is collapsing and meaningless. Thus, in my opinion, a systematic evaluation of explainability is essential in papers claiming explainability. Without this assessment, I do not think that the paper's claims about the faithfulness of explanations (especially those provided by CBM) are valid. Some datasets, such as CUB [f], provide ground-truth concept labels (attributes) and captions [g], making it possible to perform a quantitative evaluation of the concept. If it is still difficult to perform a mechanical quantitative evaluation, it is possible to introduce a human evaluation, as is done in [d].
> >
> > [e] Yamaguchi, Shin'ya, and Kosuke Nishida. "Explanation Bottleneck Models." AAAI 2025.
> >
> > [f] Wah, Catherine, et al. "The caltech-ucsd birds-200-2011 dataset." (2011).
> >
> > [g] Reed, Scott, et al. "Learning deep representations of fine-grained visual descriptions." CVPR 2016.

---

> > > ### Author Response · Authors · 2025-04-03
> > >
> > > We thank you for the follow-up.
> > >
> > > > If there is no baseline to compare and no technical novelty, then this is simply the result of a naive implementation, and efficiency is hardly the main contribution of the paper.
> > >
> > > The technical novelty is the adapter trained to bridge the image and text space, thus making it applicable to **arbitrary vision backbones**, not just models with a CLIP backbone. Specifically, in contrast to existing work, we propose to align the computed cosine similarities from our adapter for classes, with the original softmax distribution, in order **to be faithful to the original classifier’s distribution** and thus the original model. Reviewer HxVm called this “a clever way to get around having to use labeled data and become specific to the architectures in use”.
> > >
> > > Indeed, as we mentioned before, the efficiency **is not the main contribution of the paper**. Rather, it is the 1) faithfulness to the original classifier through our novel modeling explained above, and  2) being label-free, which makes it “both practical (requiring minimal data annotation) and broadly applicable to various existing visual classifiers” (reviewer 4kqE).
> > >
> > > Regarding comparisons to baselines: We **do compare** against **5** baselines across different architectures **including CNNs and transformers** on the main downstream task of CBMs and do compare against the two recent works on the downstream task of text decoding.  Our method achieves SoA results and all in a zero-shot setting. We do not compare to Text-to-Concept work [a] as they do not report any quantitative results on CBMs, only showing one qualitative example on CBMs (Figure 7 in their paper). CBMs are one of the primary downstream applications of text labeling for vision models.
> > >
> > > > What is the difference between the explanation given by CLIP and that given by the proposed method?
> > >
> > > The goal of the research is to make text accessible for *any* visual classifier, as currently this is restricted to CLIP models. The explanations we therefore get reflect the classifier (e.g., DINO) rather than CLIP. Now why is that even important? Because we may wish to explain a specific classifier and understand the biases and shortcuts it learns, through text. For example, one classifier could pay attention to shape and texture cues whereas the CLIP embedding could focus on color and “internet-popular” descriptions due to the training data of CLIP. Different models offer different insights to understand. If we can only produce text explanations for CLIP (or using CLIP, as in [a]), we lose the ability to interpret specialized or fine-tuned models—where interpretability often matters most (e.g., in medical imaging). There are many other text-based interpretability applications we could perform with our method, as well as zero-shot vision-language applications that were previously limited to CLIP.
> > >
> > > > The limited generalization in fine-grained recognition is critical
> > >
> > > Please note that this is not a problem with our formulation. This is the problem of the classifier itself. If the classifier (e.g., ResNet) cannot generalize to OOD fine-grained classes (which it does not), then **we cannot expect** our text-transformed classifier to do so. This is because our transformed classifier is a **replication** of the original classifier, and therefore inherits all its limitations. In fact, it would be bad to expect our transformed classifier to perform better than the original classifier because this means that it does not reflect its original reasoning. Also as shown in Table 5 in the Supp., better classifiers achieve better OOD fine-grained results, which perfectly aligns with the trend in OOD literature, as shown in the blue line in Figure 1 (top left) in the work of [R1], which represents OOD generalization of ImageNet-trained classifiers.
> > >
> > > [R1] Robust fine-tuning of zero-shot models
> > >
> > > > The example you presented assumes a one-to-one correspondence between visual class objects and text, but text data is flexible enough to represent the relationship between classes.
> > >
> > > The example we gave (feature-to-feature, one-to-one correspondence) was highlighting the limitations of the feature alignment loss that you suggested. We mentioned that this way **does not** represent the relationships between classes and the classifier’s way of reasoning over them. We specifically wrote in our rebuttal: “Using a one-to-one feature-to-feature mapping as the reviewer suggests will ignore the relation of other classes to the image, and will ignore the classifier’s reasoning process. Therefore, using distribution alignment loss cares about the distribution of the image over *all classes*, which is also how the classifier reasons and makes decisions”. By *all classes*, we mean relationships between classes, which is achieved by the softmax over all classes with respect to the image in our loss function. In conclusion, the example we gave was to demonstrate the limitations of your suggested counter approach.

---

### Official Review · Reviewer_qJGr · 2025-03-13

**Overall Recommendation:** 3

**Summary:**

The paper introduces a method to provide text explanations for vision models. Given textual representations of the image classes, a text encoder, and a visual classifier to interpret, the authors method associate image samples to textual classes by essentially training an MLP to calibrate the vision latent space to the text encoder latent space. After training, this MLP can be used to generate text explanations via "text queries". They demonstrate this by generating textual concepts using CBMs, and by generating a natural language explanation using  a pretrained language model. Crucially, the method is zero-shot in the sense that it does not need to be trained on paired samples (images, textual explanations), providing advantages to models such as CLIP. The authors demonstrate their method on an extensive set of 40 visual classifiers.

**Claims And Evidence:**

I find the paper and the method introduced intuitive, novel to the best of my knowledge and evidence sufficiently compelling for the applications shown. And overall I do like very much the idea of generating faithful text explanations for a pure vision model. I do however have some concerns which I will line out in the remainder of this section and in the next sections.
1) The authors repeat that their approach is faithful. Their experiments on models trained on ImageNet (table 1) are convincing for a first assessment. However, I am wondering if this faithfulness always hold. Some scenarios to consider:
    - How does the faithfulness vary as a function of the number predicted classes and the text encoder used?
    - How does the faithfulness vary as a function of the performance of the to-be-explained classifier?
    - How does faithfulness vary as a function of the MLP?

The reasons I am asking is that I assume that faithfulness depends on how well the approximation $W \approx VU^T$ holds, with $W$ and $U$ corresponding to the notation introduced by the authors in Section 3, and $V$ somewhat related to $f,\tilde{f}, MLP$. Would be interesting if the authors could comment in this direction, since faithfulness is one of the main and most repeated claims, but I do not see it addressed in the limitations in the supplementary.

2) Along the lines of faithfulness, now in the context of CBMs (first application), towards the end of section 4.1 the authors state that their CBM derivation is faithful because they do not change $U$. However, this seems wrong to me. Faithfulness of the "CBM transformation" evidently depends on the concepts in C, no? specifically it will probably depend on how the selected concepts are correlated to each other, as well as how they are/they are not correlated to the classes of the vision task. Am I missing something?

**Essential References Not Discussed:**

None to my knowledge.

**Experimental Designs Or Analyses:**

1) I personally find the qualitative examples (e.g. Figure 5) not particularly convincing. The authors interpret the results with statements such as (L. 322) "we see that the image is predicted as a “goose” because it has duck-like features". However, I find these findings a lilttle bit left to the human interpretation. More specifically, how do we actually know that the classifier actually classifies the image as goose because of so-called duck-like features? What if it is because in all the training dataset pictures all the ducks are on the grass, and the "duck" concept is activate because of the greenery in the image? Furthermore, from an interpretability perspective, simply stating "duck-like features" as an explanation may not be enough. An ornithologist might want to know which specific duck-like features are used by the model. The authors should perhaps comment on this and possible add it as a limitation of their method.
2) Following the point above, I think the CBM use case and experimental results could be strengthened with human user studies.
3) I find the experiments for the second application "Zero-Shot Decoding of Visual Features into Text" not really towards the goal stated by the authors (generate text to explain a visual model). It seems to me that the experiments confound the merits of the authors method, with the generalization capability of the underlying visual model and of the pretrained LLM to the COCO dataset. Specifically, table 3 simply shows to me that ConvNext together with LLM, and the authors construct, generate captions that correlate to the ground truth captions better than the CLIP-based models. However, it does not tell me anything regarding whether a human-user would understand how the visual model itself "reasons" about an image to achieve a prediction. Again, I think a human-user study could be a good way (although not the most prompt) to evaluate if their method enables interpretability for visual models.

**Methods And Evaluation Criteria:**

I find the method simple and reasonable for the goal stated by the authors (making a visual model text-queryable to generate explanations). However, I think faithfulness should be more thoroughly assessed and discussed (see above).

**Other Comments Or Suggestions:**

No other major comments.

**Other Strengths And Weaknesses:**

Paper if fairly easy to read. I think the idea of generating textual explanation for a vision model with no ground-truth annotations is fairly unexplored in literature, so I appreciate the originality and the simplicity of the idea. I think better experiments would strengthen the paper.

**Questions For Authors:**

No further major questions.

**Relation To Broader Scientific Literature:**

The authors provide a post-hoc global explainability method for image classification models. Furthermore, among post-hoc methods, I would say that the proposed method belongs to the class of surrogate methods (e.g. LIME) which attempt to approximate the original model in order to remain faithful (in this case an MLP is trained for "cross-modality calibration"). I believe the idea is simple,yet not quite explored in the literature to the best of my knowledge.

**Theoretical Claims:**

No theoretical claims are stated.

---

> ### Author Rebuttal · Authors · 2025-03-30
>
> We thank you for your time and effort in reviewing our paper, and for the valuable feedback. We are glad that you liked our paper and thank you for all the positive points you reported for our work. Below, we address your concerns.
>
> > How does the faithfulness vary?
>
> As a function of different text encoders, we showed in Table 6 of the appendix that the performance is not affected by different text encoders. The difference is at max 0.07 points for any text encoder we use. We also justified the reason for this in L703.
>
> We also test the impact of the MLP on the faithfulness through the following ablations:
>
> - Mean Ablation: For an image, we replace the input features to the MLP with a constant mean of the features calculated across the full ImageNet validation set.
> - Random Features: For an image, we replace the input features to the MLP with random values sampled from a normal distribution with a mean and standard deviation equal to that of the features calculated across the full ImageNet validation set
> - Random Weight Ablation: We randomize the weights of the MLP projection
> - Shuffled Ablation: For an image, we replace the input features to the MLP with input features of another random image in the validation dataset
>
> For all ablations, we calculate the ImageNet validation accuracy. We expect the accuracy percentage to drop in all ablations. This is clearly shown in the Table below. The accuracy drops to near zero in the ablations.
>
> |Model|Mean Feature|Random Features|Shuffled Features|Random Weights|
> |-|-|-|-|-|
> |ResNet101v2|Ours: 81.49/Ablated: 0.10 |Ours: 81.49/Ablated: 0.11|Ours: 81.49/Ablated: 1.70 |Ours: 81.49/Ablated: 0.11|
> |ConvNeXt-Base|Ours: 83.88/Ablated: 0.10|Ours: 83.88/Ablated: 0.11|Ours: 83.88/Ablated: 1.79|Ours:83.88/Ablated: 0.10|
> |BeiT-L/16| Ours: 87.22/Ablated: 0.10|Ours: 87.22/Ablated: 0.11|Ours: 87.22/Ablated: 1.87|Ours: 87.22/Ablated: 0.11|
> |DINOv2-B| Ours: 84.40/Ablated:0.10| Ours: 84.40/Ablated: 0.13| Ours: 84.40/Ablated: 1.76|Ours: 84.40/Ablated: 0.09|
>
> > Faithfulness of the CBM transformation depends on the concepts
>
> Our Zero-Shot CBM formulation is general and flexibly allows to use any concept set at test time directly (on-the-fly), including future improvements on reference concept sets. Indeed, if a (very) poor concept set is used, the approach is not able to faithfully reconstruct the original distribution. We assume that for the CBM the provided concept set is sufficiently expressive, which is the case even for most automatically generated concept sets such as from a LLM or the top 20k most common words. In this work, we chose the 20K most common words in English as our concept test for the sole reason of establishing a fair comparison with other works. We will address the case of poor choices for concept sets in our extended limitation discussion.
>
> > I find the qualitative examples a little bit left to the human interpretation.
>
> While we acknowledge that our qualitative example in Figure 5 may appear open to human interpretation and subject to confounding factors (e.g., background cues like greenery), this observation directly reflects the inherent ambiguity of the concept set. As mentioned in the previous concern, our zero-shot CBM formulation (Eq. 2) is independent of the concept set used (we do not train on any concept set), allowing for diverse input concept sets that can be used at test time directly (on-the-fly) that can yield more precise, attribute-specific interpretations; we refer you to C1 of Reviewer 4kqE for further clarification and qualitative examples on different concept sets attached in an anonymous link.
>
> > Table 3 simply shows to me that ConvNext together with LLM, and the authors construct, generate captions that correlate to the ground truth captions better than the CLIP-based models. However, it does not tell me anything regarding whether a human-user would understand how the visual model itself "reasons" about an image to achieve a prediction.
>
> The table indeed shows that ConvNext (and some other models ) with our adaptation performs better than the widely-used CLIP-based models. We agree that a user-centric benchmark would be very valuable for evaluation, which however does not exist yet. The main issue is, that we don't know what the model uses for reasoning (no ground truth), so even if a human user study would be conducted, the user could only judge whether the generated text is sensible with respect to the input and corresponding predicted output. This hence does not evaluate whether the textually-decoded visual features correspond to the network's inner reasoning. There is currently a gap in the literature regarding a proper benchmark for such models. We have a corresponding discussion of the limitation of this evaluation in L355 (left) - L359 in the main manuscript. As there is no ground-truth of what the model uses for reasoning, using annotated captions is the best approximation we can have.

---

### Official Review · Reviewer_4kqE · 2025-03-14

**Overall Recommendation:** 3

**Summary:**

The authors propose a method to convert a pre-trained visual classifier into a text-based classifier that supports interpretability through natural language. Specifically, they train an MLP layer to map visual features (extracted by an existing visual classifier) into a text-embedding space, where the cosine similarity with class names reflects the original classifier’s output probabilities. This approach requires no additional annotated data. The paper demonstrates that this method can effectively produce zero-shot concept bottleneck models (CBMs), as well as zero-shot decoding of the visual features into text descriptions.

## Update after rebuttal
My initial concern has been somewhat addressed through rebuttal; however, I agree with the other reviewer’s concern regarding faithfulness, which I believe was not fully resolved in the rebuttal. Therefore, I am maintaining my original score.

**Claims And Evidence:**

* The authors assert that their technique can transform any existing visual classifier into a CBM-like model. They provide a mathematical formulation in Equation (2) to justify why the resulting classifier can be interpreted under the CBM framework.

* In terms of performance, they show that their approach preserves much of the original classifier’s accuracy. Through experiments on ImageNet-1K with various architectures (e.g., ViT, CNN-based models), they report minimal drops in accuracy compared to the unmodified baseline classifiers.

**Essential References Not Discussed:**

There appear to be no critical references missing from the paper’s discussion.

**Experimental Designs Or Analyses:**

* The experiments focus on validating that the modified classifier behaves similarly to the original, with minimal accuracy drop on ImageNet-1K.
* By mapping visual features into a text space, they also showcase zero-shot text decoding. While the authors provide examples illustrating how the method can generate text from visual features, it remains an open question whether the generated text always corresponds to truly “new” concepts or simply re-labels existing classes in a less structured manner.

**Methods And Evaluation Criteria:**

* The authors use the ImageNet-1K dataset, a standard benchmark for image classification, to assess their approach across multiple architectures (ViT, CNN, etc.).

* They compare top-1 classification accuracy before and after the transformation. Additionally, they evaluate zero-shot concept bottleneck capability and text decoding performance to demonstrate the broader utility of their method.

**Other Comments Or Suggestions:**

No

**Other Strengths And Weaknesses:**

**Strengths:**

1. The proposed zero-shot CBM is both practical (requiring minimal data annotation) and broadly applicable to various existing visual classifiers.
2. The experimental results show only minor performance degradation, which makes the method attractive for real-world adoption.
3. The zero-shot decoding of visual features into text introduces new possibilities for model interpretability.

**Weaknesses:**
1. In the examples (e.g., Figure 6), the “concepts” often appear more like class labels rather than truly granular attributes, raising the question of how meaningful these concepts are in practice.
2. CBMs typically require that each concept is a distinct factor contributing to the final prediction. However, the paper’s examples sometimes resemble re-labeled classes rather than a deeper set of human-recognizable attributes. It’s unclear if this fully aligns with the spirit of CBMs, which aim to make interpretable classification process based on concepts that is deeply related to classes.

**Questions For Authors:**

* While Equation (2) demonstrates that the concept space and class space can be projected and aligned, how does one ensure that the discovered “concepts” are truly acting as the justification or rationale for the classification rather than merely providing similar or re-labeled classes?
* How does the paper ensure that these text-based “concepts” fulfill the CBM philosophy of providing a transparent, concept-level explanation for the final class decision?

**Relation To Broader Scientific Literature:**

* The paper relates to various lines of research aiming to interpret visual classifiers via textual or concept-based explanations, such as network dissection, concept discovery, and concept bottleneck models.
* Traditional classification models trained on discrete class labels give limited insight into whether the learned features encapsulate more general attributes or “concepts.” This work attempts to shed light on those features by projecting them into a text space, thereby re-purposing them to be more interpretable.

**Theoretical Claims:**

The paper does not provide a formal proof or convergence analysis, aside from referencing Equation (2), which explains how the learned MLP mapping can be interpreted within a CBM-like framework. Beyond that, there are no detailed theoretical proofs.

---

> ### Author Rebuttal · Authors · 2025-03-30
>
> We thank you for your time and effort in reviewing our paper and for the valuable feedback. We thank you also for acknowledging the strengths of our paper. In what follows, we address your concerns and remarks.
>
> > [C1] In the examples, the “concepts” often appear more like class labels rather than granular attributes, raising the question of how meaningful these concepts are in practice.... the paper’s examples sometimes resemble re-labeled classes rather than a set of attributes.….How does one ensure that the discovered “concepts” are not merely providing similar or re-labeled classes?
>
> This issue is due to the concept set we used, it is not an issue of our formulation. Unlike existing CBM works that train on a specific concept set, our zero-shot CBMs allow us to use any concept set at test time directly (on the fly); this is mentioned in L197 (left). Our method is completely independent of the concept set, and we do not train our core method (Section 3) nor our CBMs (Section 4.1) on a concept set (as mentioned in L127 (right) and detailed in Section F of the appendix). The chosen concept set is merely used at test time in the applications, and our method is flexible to any concept set as it merely involves encoding the concept set with the text encoder. In the example in Figure 5, we used the 20K most common words in English as our concept set. The 20K concept set includes both class-like names (e.g., duck, bird, pigeon) as well as granular attributes, and the reason for using it in our work as the main concept set is to establish a fair comparison with other works that also used it [R1].
>
> In Figure 5, as mentioned in the Figure caption, while the last three examples use the 20K concept set, the first example uses LF-ImageNet—an attribute-specific concept set extracted from ImageNet classes using a large language model. We include examples from different sets precisely to highlight the flexibility of our approach to different concept sets. When using the 20K-word set, class-like names often appear closer to the image in the embedding space than fine-grained attributes, which explains why they dominate the top detected concepts. In Figure 6, we again use the LF-ImageNet set, which focuses on fine-grained attributes (e.g., powerful jaws, harpoon, gills, large triangular fins, etc.).
>
> To address your concern even better, we show in an anonymous link the top-detected concepts for the second images (goose) shown in Figure 5, using exactly the same setting but with various different concept sets. We used the following concept sets:
>
> - Creating our own concept set specific to attributes from ImageNet by prompting GPT4o-mini with the prompt: “Which physical features and attributes make a {class name} different from others of the same type?”, where {class name} is a class name from ImageNet.
> - The concept set of LF-ImageNet
> - Creating our own concept set specific to attributes from ImageNet by prompting GPT4o and Llama LLMs with the prompt: “What are useful visual features for distinguishing a {class name} in a photo?”, where {class name} is a class name from ImageNet.
>
> The anonymous link is here: https://github.com/user-attachments/assets/cd8e0ba0-062b-4d44-a541-2fda87f870a2
>
> As it can be shown, when using a concept set of fine-grained attributes, this issue no longer exists. Again, we merely used the 20K as our main concept set in order to establish a fair comparison with other works.
>
> > [C2] How does the paper ensure that these text-based “concepts” fulfill the CBM philosophy of providing a transparent, concept-level explanation for the final class decision?
>
> Indeed, the concepts should provide a transparent explanation for the final classification. Our designed approach follows this principle, where the classification layer is a simple linear layer from the concepts to the output (classes):
>
> $ pred = w_1 c_1 + w_2 c_2 + w_3 c_3 + \dots + w_Z c_Z $, where $c$ is the concept, $w$ is the weight connecting the concept to the prediction class, and $Z$ is the total number of concepts. As this is a linear layer, we can interpret the classification decision by looking at the concepts with the highest weights connecting to the prediction. However, by merely looking at the weights, we only get a global explanation (since the weights are constant and do not change as a function of the input). That is why, in order to make the explanation local to the prediction, we have to multiply those weights with their concept activation scores obtained by feeding the image (as mentioned in L320-left). Therefore, we dissect the prediction into its input elements: $ pred = [w_1 * c_1 , w_2 * c_2 , w_3 * c_3 , \dots , w_Z * c_Z] $ and then obtain the top-k elements. The numbers shown next to the concepts in Figure 5 are calculated by multiplying the weight of the concept by its concept activation score ($w_x * c_x$).
>
> [R1] DN-CBM: Discover-then-Name: Task-Agnostic Concept Bottlenecks via Automated Concept Discovery, ECCV 2024

---

> > ### Comment · Reviewer_4kqE · 2025-04-07
> >
> > Thank you for your response. However, I feel that the reply does not fully address the question I raised.
> >
> > > This issue is due to the concept set we used, it is not an issue of our formulation.
> >
> > I also agree that this may not be a problem with the proposed formulation itself. However, if the concept set already includes concepts that are closely aligned with the target classes, and if class predictions are effectively determined during the concept prediction stage and directly carried over to the final class prediction, I’m not sure this can be considered a truly new framework for interpretability using CBMs.
> >
> > Since empirical validation of the proposed framework is also crucial, I believe that experiments should have been designed with this concern in mind.
> >
> > Furthermore, the authors highlight that one of the strengths of the proposed ZS-CBM is that it preserves performance when converting existing models into the CBM structure. But if this performance preservation is due to the reasons mentioned above, can we truly consider this a meaningful transition to a CBM in the sense that we aim for with interpretable models?

---

> > > ### Author Response · Authors · 2025-04-08
> > >
> > > We thank the reviewer for the valuable comment. We address your concern below:
> > >
> > > To ensure that the concepts are meaningful and free of terms that are overly similar or directly derived from the target classes, we apply a rigorous filtering procedure to the concept set. Specifically, we remove any terms that 1) exactly match the target class name, 2) any constituent words that form the class name (for example, eliminating “tiger” and “shark” when the class name is “tiger shark”), 3) terms corresponding to the subparent class (e.g., “fish” for the class “tiger shark”), 4) terms corresponding to the parent class (e.g., “animal” for the class “tiger shark”), 5) other species within the same category, and 6) any synonyms of the target class name. We obtain 3,4,5 and 6 with an LLM for scalability (we used ChatGPT4o-mini). We perform this filtering procedure on the concept set before we apply our ZS-CBM formulation (Eq. 2). We test a selection of models and present the results below. As seen, the performance is completely independent of these terms. The accuracy remains the same on all models tested.
> > >
> > >
> > > |Model|original set|filtered set|
> > > |-|-|-|
> > > |ResNet50|73.9|73.9|
> > > |ResNet50v2|78.1|78.1|
> > > |ResNet101|75.3|75.3|
> > > |ViT-B/16|79.3|79.3|
> > > |ViT-B/32|73.3|73.3|
> > > |BeiT-B/16|83.0|83.0|
> > > |BeiT-L/16|86.2|86.2|
> > > |ConvNeXt-Base|84.0|84.0|
> > > |ConvNeXtV2-Bpt@384|86.3|86.4|

---

### Decision · Program_Chairs · 2025-05-01

**Decision:**

Reject

**Comment:**

In this work, the authors present a method for transforming image classifier into a text-based classifier that supports interpretability through natural language prompts. The authors train a network to map image features into a text-embedding space such that the network may be used to identify text explanations for a classification. Critically, their method does not require paired samples as is required for standard image-text models, e.g. CLIP. The authors demonstrate their method on an extensive set of 40+ image classification architectures to showcase the generality of their approach. Some reviewers were impressed that the method appeared to generally increase model interpretability without requiring additional labeled training data. However, several reviewers raise concerns about the reliability and faithfulness of the explanations provided by the method. In particular, some reviewers were not convinced by the quantitative evaluations. Two substantive reasons for this is that (1) task accuracy is not appropriate for evaluating explanations and (2) the lack of compelling evidence on fine-grained classification. During subsequent reviewer discussion, the most positive reviewer notably lowered their score and concurred with some of the concerns raised by other reviewers. Upon my review, I share the same central concerns about the reliability and faithfulness of the explanations provided by the method. Although a compelling and novel method, this paper will not be accepted to this conference due to this unresolved concern. The authors are encouraged to pursue fine-grained classification in their future experiments and focus on demonstrating more convincing quantitative evaluations methods.